# ATISS: Autoregressive Transformers for Indoor Scene Synthesis

**Despoina Paschalidou**[*,1,3,4]    **Amlan Kar**[4,5,6]    **Maria Shugrina**[4]    **Karsten Kreis**[4]

**Andreas Geiger**[1,2,3]    **Sanja Fidler**[4,5,6]

[1]Max Planck Institute for Intelligent Systems Tübingen    [2]University of Tübingen
[3]Max Planck ETH Center for Learning Systems
[4]NVIDIA    [5]University of Toronto    [6]Vector Institute
{firstname.lastname}@tue.mpg.de    {amlank, mshugrina, kkreis, sfidler}@nvidia.com

## Abstract

The ability to synthesize realistic and diverse indoor furniture layouts automatically or based on partial input, unlocks many applications, from better interactive 3D tools to data synthesis for training and simulation. In this paper, we present ATISS, a novel autoregressive transformer architecture for creating diverse and plausible synthetic indoor environments, given only the room type and its floor plan. In contrast to prior work, which poses scene synthesis as sequence generation, our model generates rooms as unordered sets of objects. We argue that this formulation is more natural, as it makes ATISS generally useful beyond fully automatic room layout synthesis. For example, the same trained model can be used in interactive applications for general scene completion, partial room re-arrangement with any objects specified by the user, as well as object suggestions for any partial room. To enable this, our model leverages the permutation equivariance of the transformer when conditioning on the partial scene, and is trained to be permutation-invariant across object orderings. Our model is trained end-to-end as an autoregressive generative model using only labeled 3D bounding boxes as supervision. Evaluations on four room types in the 3D-FRONT dataset demonstrate that our model consistently generates plausible room layouts that are more realistic than existing methods. In addition, it has fewer parameters, is simpler to implement and train and runs up to 8x faster than existing methods.

## 1  Introduction

Generating synthetic 3D content that is both realistic and diverse is a long-standing problem in computer vision and graphics. In the last decade, there has been increased demand for tools that automate the creation of 3D artificial environments for applications like video games and AR/VR, as well as general 3D content creation [61, 16, 36, 4, 62]. These tools can also synthesize data to train computer vision models, avoiding expensive and laborious annotations. Generative models [28, 19, 13, 29, 56] have demonstrated impressive results on synthesizing photorealistic images [7, 1, 24, 8, 25] and intelligible text [46, 2], and are beginning to be adopted for the generation of 3D environments.

Recent works proposed to solve the scene synthesis task by incorporating procedural modeling techniques [45, 43, 23, 9] or by generating scene graphs with generative models [34, 57, 65, 35, 44, 64, 63, 27, 12]. Procedural modeling requires specifying a set of rules for the scene formation process,

---

[*]Work done during Despoina's internship at NVIDIA.

35th Conference on Neural Information Processing Systems (NeurIPS 2021).

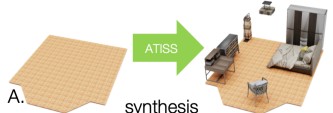 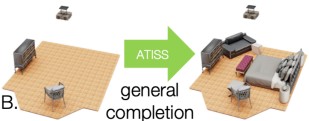 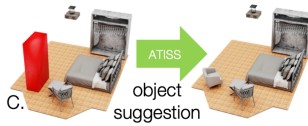

**Figure 1: Motivation** In addition to fully automatic layout synthesis (A), our formulation in terms of unordered sets of objects allows our model to be used for novel interactive applications with versatile user control: scene completion given any number of existing furniture pieces of any class pinned to a specific location by the user (B), and object suggestions with user-provided constraints (object centroid constraint shown in red) (C).

but acquiring these rules is a time-consuming task, requiring skills of experienced artists. Similarly, graph-based approaches require scene graph annotations, which may be laborious to obtain.

Another line of research utilizes CNN-based [58, 47] and transformer-based [59] architectures to generate rooms by autoregressively selecting and placing objects in a scene, i.e. one after the other. These approaches represent scenes as ordered sequences of objects. Typically, the ordering is defined using the spatial arrangement of objects in a room (e.g. left-to-right) [22] or the object class frequency (e.g. most to least probable) [47, 59]. Such orderings impose unnatural constraints on the scene generation process, inhibiting practical applications. For example, in [47, 59], which order objects by class frequency, the probability of a bed (more common) appearing after an ottoman (less common) in the training set is zero. As a result, these methods cannot generate more common objects after less common objects, which makes them impractical for interactive tasks like general room completion and partial room re-arrangement, where input is unconstrained (e.g. Fig.1B).

To address these limitations, we pose scene synthesis as an unordered set generation problem and introduce ATISS, a novel autoregressive transformer architecture to model this process. Given a room type (e.g. bedroom, living room) and its shape, our model generates meaningful furniture arrangements by sequentially placing objects in a permutation-invariant fashion. We train ATISS to maximize the log-likelihood of all possible permutations of object arrangements in a collection of training scenes, labeled only with object classes and $3D$ bounding boxes, which are easier to obtain, than costly support relationship [57] or scene graph annotations [34]. Unlike existing works [58, 47, 59], we propose the first model to perform scene synthesis as an autoregressive *set generation* task. ATISS is significantly simpler to implement and train, requires fewer parameters and is up to $8\times$ faster at run-time than the fastest available baseline [59]. Furthermore, we demonstrate that our model generates more plausible object arrangments without any post-processing on the predicted layout. Our formulation allows applying a single trained model to automatic layout synthesis and to a number of interactive scenarios with versatile user input (Fig.1), such as automatic placement of user-provided objects, object suggestion with user-provided constraints, and room completion. Code and data are publicaly available at https://nv-tlabs.github.io/ATISS.

## 2 Related Work

In this section, we discuss the most relevant literature on interior scene synthesis, as well as transformer architectures [56] in the context of generative modeling.

**Procedural Modeling with Grammars:** Procedural modeling describes methods that recursively apply a set of functions for content synthesis. Grammars are a formal instantiation of this idea and have been used for modeling 3D structures such as plants [52], buildings and cities [37, 40], indoor [45] and outdoor [43] scenes. [52] employed reversible-jump MCMC to control the output of stochastic context-free grammars. Meta-Sim [23] learned a model that modifies attributes of scene graphs sampled from a known probabilistic context-free grammar to match visual statistics between generated and real data. [9] extended this model to also learn to sample from the grammar, allowing context dependent relationships to be learnt. Concurrently, [44] employed Grammar-VAE [31] to generate scenes using a scene grammar generated from annotated data. In contrast, our model implicitly encapsulates inter-object relationships, without having to impose hand-crafted constraints.

**Graph-based Scene Synthesis:** Representing scenes as graphs has been extensively studied in literature [34, 57, 65, 35, 44, 64, 63, 27, 12]. Zhou et al. [65] introduced a neural message passing algorithm for scene graphs that predicts the category of the next object to be placed at a specific location. Similarly, [34, 64, 44, 35] utilized a VAE [28] to synthesize 3D scenes as parse trees [44], adjacency matrices [64], scene graphs [35] and scene hierarchies [34]. Concurrently, [57, 63] adopted a two-stage generation process that disentangles planning the scene layout from instantiating the scene based on this plan. Note that graph-based models require supervision either in the form of

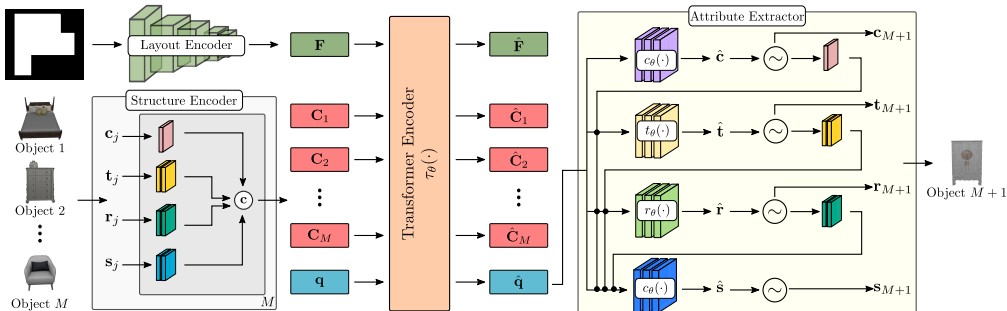

**Figure 2: Method Overview.** Starting from a scene with $M$ objects and a floor layout, the **layout encoder** maps the floor into a feature representation $\mathbf{F}$ and the **structure encoder** maps the objects into a context embedding $\mathbf{C} = \{\mathbf{C}_j\}_{j=1}^M$. The floor layout feature $\mathbf{F}$, the context embedding $\mathbf{C}$ and a learnable query vector $\mathbf{q}$ are then passed to the **transformer encoder** that predicts $\hat{\mathbf{q}}$. Using $\hat{\mathbf{q}}$ the **attribute extractor** autoregressively predicts the attribute distributions that are used to sample the attributes for the next object to be generated.

relation graphs [57, 63, 35] or scene hierarchies [34]. In contrast, ATISS infers functional and spatial relations between objects directly from data labeled only with object classes and 3D bounding boxes.

**Autoregressive Scene Synthesis:** Closely related to our work are autoregressive indoor scene generation models [58, 47, 59]. Ritchie et al. [47] introduced a CNN-based architecture that operates on a top-down image-based representation of a scene and inserts objects in it sequentially by predicting their category, location, orientation and size with separate network modules. [47] requires supervision in the form of 2D bounding boxes as well as auxiliary supervision such as depth maps and object segmentation masks. In concurrent work, Wang et al. [59] introduced SceneFormer, a series of transformers that autoregressively add objects in a scene similar to [47]. Both [47, 59] use separate models to generate object attributes (e.g. category, location) that are trained independently and represent scenes as ordered sequences of objects, ordered by the category frequency. In contrast, we propose a simpler architecture that consists of a single model trained end-to-end to predict all attributes. We provide experimental evidence that our model generates more realistic object arrangements while being significantly faster. While [47, 59] assume a fixed ordering of the objects in each scene, our model does not impose any constraint on the ordering of objects. Instead, during training, we enforce that our model generates objects with all orderings, in a permutation invariant fashion. This allows us to represent scenes as unordered sets of objects and perform various interactive tasks such as rearranging any object in a room or suggesting new objects given any room.

**Transformers for Set Generation:** Transformer models [56] demonstrated impressive results on various tasks such as machine translation [50, 39], language-modeling [2, 10], object detection [33, 3, 66], image recognition [14, 53], semantic segmentation [60] as well as on image [42, 26, 5, 15, 54] and music [11] generation tasks. While there are works [32, 30] that utilize the permutation equivariance property of transformers for unordered set processing and prediction, existing generative models with transformers assume ordered sequences [2, 5, 6] even when there exists no natural order such as for pointclouds [38] and objects in a scene [59]. Instead, we introduce an autoregressive transformer for unordered set generation that enforces that the probability of adding a new element in the set is invariant to the order of the elements already in the set. We show that for the scene synthesis task, our model outperforms transformers that consider ordered sets of elements in every metric.

## 3 Method

Given an empty or a partially complete room of a specific type (e.g. bedroom) together with its shape, as a top-down orthographic projection of its floor, we want to learn a generative model that populates the room with objects, whose functional composition and spatial arrangement is plausible. To this end, we propose an autoregressive model that represents scenes as *unordered sets of objects* (Sec. 3.1) and describe our implementation using a transformer network (Sec. 3.2). Finally, we analyse the training and inference details of our method (Sec. 3.3).

### 3.1 Autoregressive Set Generation

Let $\mathcal{X} = \{\mathcal{X}_1, \ldots, \mathcal{X}_N\}$ denote a collection of scenes where each $\mathcal{X}_i = (\mathcal{O}_i, \mathbf{F}^i)$ comprises the unordered set of objects in the scene $\mathcal{O}_i = \{o_j^i\}_{j=1}^M$ and its floor layout $\mathbf{F}^i$. To compute the likelihood

of generating $\mathcal{O}_i$ we need to accumulate the likelihood of generating $\{o_j^i\}_{j=1}^M$ autoregressively in *any order*. This is formally written as

$$p_\theta(\mathcal{O}_i|\mathbf{F}^i) = \sum_{\hat{\mathcal{O}} \in \pi(\mathcal{O}_i)} \prod_{j \in \hat{\mathcal{O}}} p_\theta(o_j^i \mid o_{<j}^i, \mathbf{F}^i), \tag{1}$$

where $p_\theta(o_j^i \mid o_{<j}^i, \mathbf{F}^i)$ is the probability of the $j$-th object, conditioned on the previously generated objects and the floor layout, and $\pi(\cdot)$ is a permutation function that computes the set of permutations of all objects in the scene. As a result, the log-likelihood of the whole collection $\mathcal{X}$ is

$$\log p_\theta(\mathcal{X}) = \sum_{i=1}^N \log \left( \sum_{\hat{\mathcal{O}} \in \pi(\mathcal{O}_i)} \prod_{j \in \hat{\mathcal{O}}} p_\theta(o_j^i \mid o_{<j}^i, \mathbf{F}^i) \right). \tag{2}$$

However, training our generative model to maximize the log-likelihood of (2) poses two problems: (a) the summation over all permutations is intractable and (b) (2) does not ensure that all orderings will have high probability. The second problem is crucial because we want our generative model to be able to complete *any partial set* in a plausible way, namely we want any generation order to have high probability. To this end, instead of maximizing (2), we maximize the likelihood of generating a scene in all possible orderings, $\hat{p}_\theta(\cdot)$, which is defined as

$$\log \hat{p}_\theta(\mathcal{X}) = \sum_{i=1}^N \log \left( \prod_{\hat{\mathcal{O}} \in \pi(\mathcal{O}_i)} \prod_{j \in \hat{\mathcal{O}}} p_\theta(o_j^i \mid o_{<j}^i, \mathbf{F}^i) \right) = \sum_{i=1}^N \sum_{\hat{\mathcal{O}} \in \pi(\mathcal{O}_i)} \sum_{j \in \hat{\mathcal{O}}} \log p_\theta(o_j^i \mid o_{<j}^i, \mathbf{F}^i). \tag{3}$$

Note that training our generative model with (3) allows us to approximate the summation over all permutations using Monte Carlo sampling thus solving both problems of (2).

**Modelling Object Attributes:** We represent objects in a scene as labeled 3D bounding boxes and model them with four random variables that describe their category, size, orientation and location, $o_j = \{\mathbf{c}_j, \mathbf{s}_j, \mathbf{t}_j, \mathbf{r}_j\}$. The category $\mathbf{c}_j$ is modeled using a categorical variable over the total number of object categories $C$ in the dataset. For the size $\mathbf{s}_j \in \mathbb{R}^3$, the location $\mathbf{t}_j \in \mathbb{R}^3$ and the orientation $\mathbf{r}_j \in \mathbb{R}^1$, we follow [48, 55] and model them with mixture of logistics distributions

$$\mathbf{s}_j \sim \sum_{k=1}^K \pi_k^s \text{logistic}(\mu_k^s, \sigma_k^s) \quad \mathbf{t}_j \sim \sum_{k=1}^K \pi_k^t \text{logistic}(\mu_k^t, \sigma_k^t) \quad \mathbf{r}_j \sim \sum_{k=1}^K \pi_k^r \text{logistic}(\mu_k^r, \sigma_k^r) \tag{4}$$

where $\pi_k^s$, $\mu_k^s$ and $\sigma_k^s$ denote the weight, mean and variance of the $k$-th logistic distribution used for modeling the size. Similarly, $\pi_k^t$, $\mu_k^t$ and $\sigma_k^t$ and $\pi_k^r$, $\mu_k^r$ ans $\sigma_k^r$ refer to the weight, mean and variance of the $k$-th logistic of the location and orientation, respectively. In our setup, the orientation is the angle of rotation around the up vector and the location is the 3D centroid of the bounding box.

Similar to prior work [47, 59], we predict the object attributes in an autoregressive manner: object category first, followed by position, orientation and size as follows:

$$p_\theta(o_j \mid o_{<j}, \mathbf{F}) = p_\theta(\mathbf{c}_j|o_{<j}, \mathbf{F})p_\theta(\mathbf{t}_j|\mathbf{c}_j, o_{<j}, \mathbf{F})p_\theta(\mathbf{r}_j|\mathbf{c}_j, \mathbf{t}_j, o_{<j}, \mathbf{F})p_\theta(\mathbf{s}_j|\mathbf{c}_j, \mathbf{t}_j, \mathbf{r}_j, o_{<j}, \mathbf{F}). \tag{5}$$

This is a natural choice, since we want our model to consider the object class before reasoning about the size and the position of an object. To avoid notation clutter, we omit the scene index $i$ from (5).

### 3.2 Network Architecture

The input to our model is a collection of scenes in the form of 3D labeled bounding boxes with their corresponding room shape. Our network consists of four main components: (i) the *layout encoder* that maps the room shape to a global feature representation $\mathbf{F}$, (ii) the *structure encoder* $h_\theta$ that maps the $M$ objects in the scene into per-object context embeddings $\mathbf{C} = \{\mathbf{C}_j\}_{j=1}^M$, (iii) the *transformer encoder* $\tau_\theta$ that takes $\mathbf{F}$, $\mathbf{C}$ and a query embedding $\mathbf{q}$ and predicts the features $\hat{\mathbf{q}}$ for the next object to be generated and (iv) the *attribute extractor* that predicts the attributes of the next object. Our model is illustrated in Fig. 2. The layout encoder is simply a ResNet-18 [20] that extracts a feature representation $\mathbf{F} \in \mathbb{R}^{64}$ for the top-down orthographic projection of the floor.

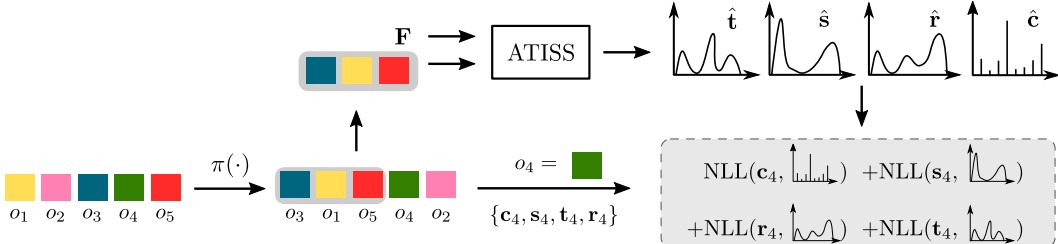

**Figure 3: Training Overview:** Given a scene with $M$ objects (coloured squares), we first randomly permute them and then keep the first $T$ objects (here $T = 3$). We task our network to predict the next object to be added in the scene given the subset of kept objects (highlighted with grey) and its floor layout feature $\mathbf{F}$. Our loss function is the negative log-likelihood (NLL) of the next object in the permuted sequence (green square).

**Structure Encoder:** The structure encoder $h_\theta$ maps the attributes of the $j$-th object into a per-object context embedding $\mathbf{C}_j$ as follows:

$$h_\theta : \mathbb{R}^C \times \mathbb{R}^3 \times \mathbb{R}^3 \times \mathbb{R}^1 \to \mathbb{R}^{L_c} \times \mathbb{R}^{L_s} \times \mathbb{R}^{L_t} \times \mathbb{R}^{L_r}$$
$$(\mathbf{c}, \mathbf{s}, \mathbf{t}, \mathbf{r}) \mapsto [\lambda(\mathbf{c}); \gamma(\mathbf{s}); \gamma(\mathbf{t}); \gamma(\mathbf{r})] \tag{6}$$

where $L_c, L_s, L_t, L_r$ are the output dimensionalities of the embeddings used to map the category, the size, the location and the orientation into a higher dimensional space respectively and $[\cdot ; \cdot]$ denotes concatenation. For the object category $\mathbf{c}_j$ we use a learnable embedding $\lambda(\cdot)$, whereas for the size $\mathbf{s}_j$, the position $\mathbf{t}_j$ and the orientation $\mathbf{r}_j$, we use the positional encoding of [56] as follows

$$\gamma(p) = (\sin(2^0 \pi p), \cos(2^0 \pi p), \dots, \sin(2^{L-1} \pi p), \cos(2^{L-1} \pi p)) \tag{7}$$

where $p$ can be any of the size, position or orientation attributes and $\gamma(\cdot)$ is applied separately in each attribute's dimension. The set of per-object context vectors synthesizes the context embedding $\mathbf{C}$ that encapsulates information for the existing objects in the scene and is used to condition the next object to be generated. Before passing the output of (6) to the transformer encoder, we map each $\mathbf{C}_j$ to 64 dimensions using a linear projection.

**Transformer Encoder:** We follow [56, 10] and implement our encoder $\tau_\theta$ as a multi-head attention transformer without any positional encoding. This allows us to learn a parametric function that computes features that are invariant to the order of $\mathbf{C}_j$ in $\mathbf{C}$. We use these features to predict the next object to be added in the scene, creating an autoregressive model. The input set of the transformer is $\mathbf{I} = \{\mathbf{F}\} \cup \{\mathbf{C}_j\}_{j=1}^M \cup \mathbf{q}$, with $M$ the number of objects in the scene. $\mathbf{q} \in \mathbb{R}^{64}$ is a learnable object query vector that allows the transformer to predict output features $\hat{\mathbf{q}} \in \mathbb{R}^{64}$ used for generating the next object to be added in the scene. The use of a query token is akin to the use of a mask embedding in Masked Language Modelling [10] or the class embedding for the Vision Transformer [49].

**Attribute Extractor:** We autoregressively predict the attributes of the next object to be added in the scene using one MLP for each attribute. More formally, the attribute extractor is defined as follows:

$$c_\theta : \mathbb{R}^{64} \to \mathbb{R}^C \qquad\qquad \hat{\mathbf{q}} \mapsto \hat{\mathbf{c}} \tag{8}$$
$$t_\theta : \mathbb{R}^{64} \times \mathbb{R}^{L_c} \to \mathbb{R}^{3 \times 3 \times K} \qquad\qquad (\hat{\mathbf{q}}, \lambda(\mathbf{c})) \mapsto \hat{\mathbf{t}} \tag{9}$$
$$r_\theta : \mathbb{R}^{64} \times \mathbb{R}^{L_c} \times \mathbb{R}^{L_t} \to \mathbb{R}^{1 \times 3 \times K} \qquad\qquad (\hat{\mathbf{q}}, \lambda(\mathbf{c}), \gamma(\mathbf{t})) \mapsto \hat{\mathbf{r}} \tag{10}$$
$$s_\theta : \mathbb{R}^{64} \times \mathbb{R}^{L_c} \times \mathbb{R}^{L_t} \times \mathbb{R}^{L_r} \to \mathbb{R}^{3 \times 3 \times K} \qquad\qquad (\hat{\mathbf{q}}, \lambda(\mathbf{c}), \gamma(\mathbf{t}), \gamma(\mathbf{r})) \mapsto \hat{\mathbf{s}} \tag{11}$$

where $\hat{\mathbf{c}}, \hat{\mathbf{s}}, \hat{\mathbf{t}}, \hat{\mathbf{r}}$ are the predicted attribute distributions and $c_\theta, t_\theta, r_\theta$ and $s_\theta$ are mappings between the latent space and the low-dimensional space of attributes. For the object category, $c_\theta$ predicts $C$ class probabilities, whereas, $t_\theta, r_\theta$ and $s_\theta$ predict the mean, variance and mixing coefficient for the $K$ logistic distributions for each attribute. To predict the object properties in an autoregressive manner, we need to condition the prediction of a property on the previously predicted properties. Thus, instead of only passing $\hat{\mathbf{q}}$ to each MLP, we concatenate it with the previously predicted attributes, mapped in a higher-dimensional space using the embeddings $\lambda(\cdot)$ and $\gamma(\cdot)$ from (6).

### 3.3 Training and Inference

During training, we choose a scene from the dataset and apply a random permutation $\pi(\cdot)$ on its $M$ objects. Then, we randomly select the first $T$ objects to compute the context embedding $\mathbf{C}$.

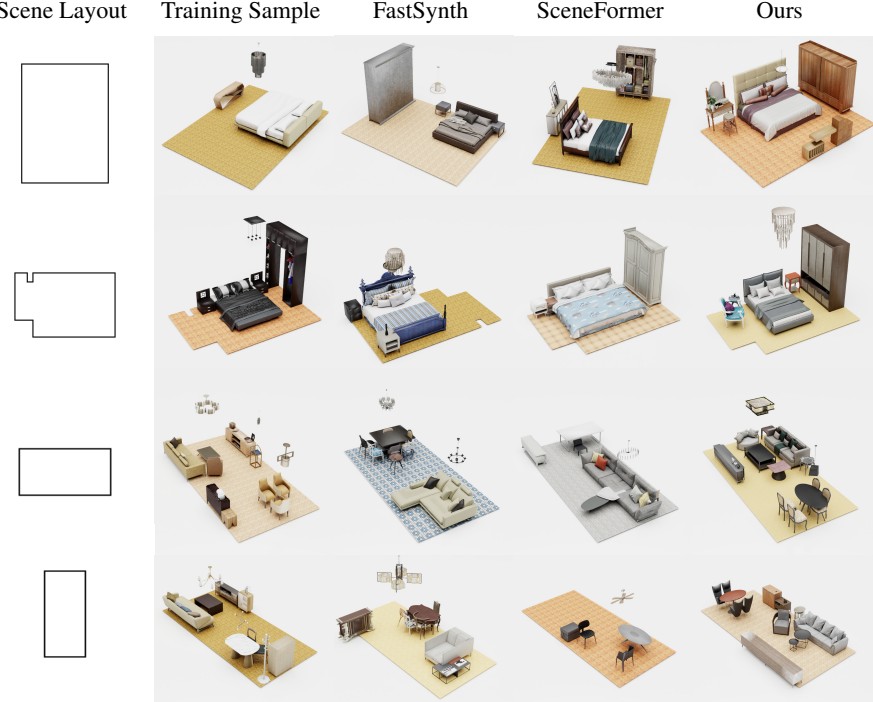

|  Scene Layout | Training Sample | FastSynth | SceneFormer | Ours |

**Figure 4: Qualitative Scene Synthesis Results**. Synthesized scenes for three room types: bedrooms (1st+2nd row), living room (3rd row), dining room (4th row) using FastSynth, SceneFormer and our method. To showcase the generalization abilities of our model we also show the closest scene from the training set (2nd column).

Conditioned on $\mathbf{C}$ and $\mathbf{F}$, our network predicts the attribute distributions of the next object to be added in the scene and is trained to maximize the log-likelihood of the $T+1$ object from the permuted scene. A pictorial representation of the training process is provided in Fig. 3. To indicate the end of sequence, we augment the $C$ object classes with an additional class, which we refer to as *end symbol*.

During inference, we start with an empty context embedding $\mathbf{C} = \emptyset$ and the floor representation $\mathbf{F}$ of the room to be populated and autoregressively sample attribute values from the predicted distributions of (8)-(11) for the next object. Once a new object is generated, it is appended to the context $\mathbf{C}$ to be used in the next step of the generation process until the *end symbol* is generated. A pictorial representation of the generation process can be found in Fig. 2. In order to transform the predicted labeled bounding boxes to 3D models we use object retrieval. In particular, we retrieve the closest object from the dataset in terms of the euclidean distance of the bounding box dimensions.

## 4 Experimental Evaluation

In this section, we provide an extensive evaluation of our method, comparing it to existing baselines. We further showcase several interactive use cases enabled by our method, not previously possible. Additional results as well as implementation details are provided in the supplementary.

**Datasets:** We train our model on the 3D-FRONT dataset [17] which contains a collection of $6,813$ houses with roughly $14,629$ rooms, populated with 3D furniture objects from the 3D-FUTURE dataset [18]. In our evaluation, we focus on four room types: (i) bedrooms, (ii) living rooms, (iii) dining rooms and (iv) libraries. After pre-processing to filter out uncommon object arrangements and rooms with unnatural sizes, we obtained 5996 bedrooms, 2962 living rooms, 2625 dining rooms and 622 libraries. We use 21 object categories for the bedrooms, 24 for the living and dining rooms and 25 for the libraries. The preprocessing steps are discussed in the supplementary.

**Baselines:** We compare our approach to FastSynth [47] and SceneFormer [59] using the authors' implementations. Note that both approaches were originally evaluated on the SUNCG dataset [51], which is now unavailable. Thus, we retrained both on 3D-FRONT. We also compare with a variant of our model that generates scenes as ordered sequences of objects (Ours+Order). To incorporate the order information to the input, we utilize a positional embedding [56] and a fixed ordering based on the object frequency as described in [59].

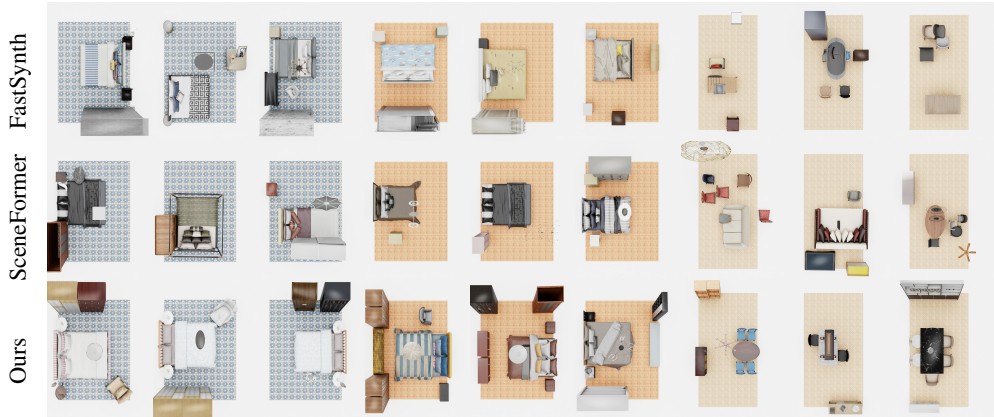

**Figure 5: Scene Diversity**. We show three generated scenes conditioned on three different floor plans for bedrooms and dining rooms. Every triplet of columns corresponds to a different floor plan.

| | FID Score (↓) | | | | Scene Classification Accuracy | | | | Category KL Divergence (↓) | | | |
|---|---|---|---|---|---|---|---|---|---|---|---|---|
| | FastSynth | SceneFormer | Ours+Order | Ours | FastSynth | SceneFormer | Ours+Order | Ours | FastSynth | SceneFormer | Ours+Order | Ours |
| Bedrooms | 40.89 | 43.17 | 38.67 | **38.39** | 0.883 | 0.945 | 0.760 | **0.562** | 0.0064 | **0.0052** | 0.0533 | 0.0085 |
| Living | 61.67 | 69.54 | 35.37 | **33.14** | 0.945 | 0.972 | 0.694 | **0.516** | 0.0176 | 0.0313 | 0.0372 | **0.0034** |
| Dining | 55.83 | 67.04 | 35.79 | **29.23** | 0.935 | 0.941 | 0.623 | **0.477** | 0.0518 | 0.0368 | 0.0278 | **0.0061** |
| Library | 37.72 | 55.34 | 35.60 | **35.24** | 0.815 | 0.880 | 0.572 | **0.521** | 0.0431 | 0.0232 | 0.0183 | **0.0098** |

**Table 1: Quantitative Comparison.** We report the FID score (↓) at $256^2$ pixels, the KL divergence (↓) between the distribution of object categories of synthesized and real scenes and the real vs. synthetic classification accuracy for all methods. Classification accuracy closer to 0.5 is better.

**Evaluation Metrics:** To measure the realism of the generated scenes, we follow prior work [47] and report the KL divergence between the object category distributions of synthesized and real scenes from the test set and the classification accuracy of a classifier trained to discriminate real from synthetic scenes. We also report the FID [21] between top-down orthographic projections of synthesized and real scenes from the test set, which we compute using [41] on $256^2$ images. We repeat the metric computation for FID and classification accuracy 10 times and report the average.

## 4.1 Scene Synthesis

We start by evaluating the performance of our model on generating plausible object configurations for various room types, conditioned on different floor plans. Fig. 4 provides a qualitative comparison of four scenes generated with our model and baselines. In some cases, both [47, 59] generate invalid room layouts with objects positioned outside room boundaries or overlapping. Instead, our model consistently synthesizes realistic object arrangements. We validate this quantitatively in Tab. 1, where we compare the generated scenes wrt. their similarity to the original data from 3D-FRONT. Synthesized scenes sampled from our model are almost indistinguishable from scenes from the test set, as indicated by the classification accuracy in Tab. 1, which is consistently around 50%. Our model also achieves lower FID scores for all room types and generates category distributions that are more faithful to the category distributions of the test set, expressed as lower KL divergence.

Scene Layout   FastSynth   SceneFormer   Ours          Scene Layout   FastSynth   SceneFormer   Ours

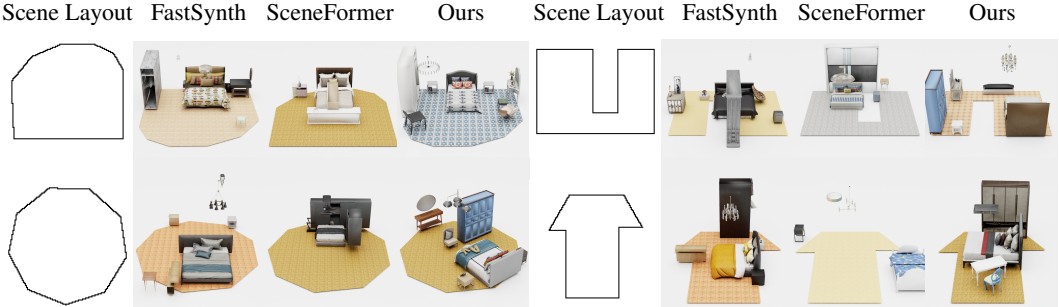

**Figure 6: Generalization Beyond Training Data**. We show four synthesized bedrooms conditioned on four room layouts that we manually designed.

| Partial Scene | FastSynth | SceneFormer | Ours+Order | Ours |
|:---:|:---:|:---:|:---:|:---:|

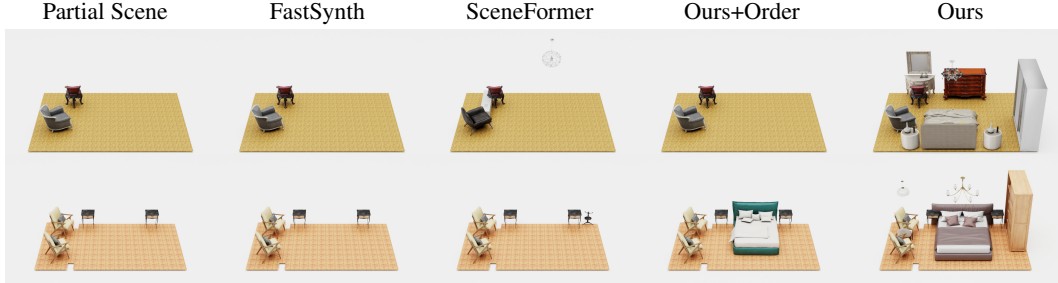

**Figure 7: Scene Completion**. Given a partial scene (left column), we visualize scene completions using our model and our baselines. Our model consistently generates plausible layouts.

|  | Bedroom | Living | Dining | Library |
|---|---|---|---|---|
| FastSynth [47] | 13193.77 | 30578.54 | 26596.08 | 10813.87 |
| SceneFormer [59] | 849.37 | 731.84 | 901.17 | 369.74 |
| Ours [47] | **102.38** | **201.59** | **201.84** | **88.24** |

**Table 2: Generation Time Comparison.** We measure time (ms) to generate a scene, conditioned on a floor plan.

| FastSynth [47] | SceneFormer [59] | Ours |
|:---:|:---:|:---:|
| 38.180 | 129.298 | 36.053 |

**Table 3: Network Parameters Comparison.** We report the number of network parameters in millions.

To showcase that our model generates diverse object arrangements we visualize 3 generated scenes conditioned on the same floor plan for all methods (Fig. 5). We observe that our generated scenes are consistently valid and contain diverse object arrangements. In comparison [47, 59] struggle to generate plausible layouts particularly for the case of living rooms and dining rooms. We hypothesize that these rooms are more challenging than bedrooms, for the baselines, due to their significantly smaller volume of training data, and the large number of constituent objects per scene (20 on average, as opposed to 8). To investigate whether our model also generates plausible layouts conditioned on floor plans with uncommon shapes that are not in the training set, we manually design unconventional floor plans (Fig. 6) and generate bedroom layouts. While both [47, 59] fail to generate valid scenes, our model synthesizes diverse object layouts that are consistent with the floor plan. Finally, we compare the computational requirements of our architecture to [47, 59]. Our model is significantly faster (Tab. 2), while having fewer parameters (Tab. 3) than both [47, 59]. Note that [47] is orders of magnitude slower because it requires rendering every individual object added in the scene.

## 4.2 Applications

In this section, we present three applications that greatly benefit by our unordered set formulation and are crucial for creating an interactive scene synthesis tool.

**Scene Completion:** Starting from a partial scene, the task is to populate the empty space in a meaningful way. Since both [47, 59] are trained on sorted sequences of objects, they first generate frequent objects (e.g. beds, wardrobes) followed by less common objects. As a result, incomplete scenes that contain less common objects cannot be correctly populated. This is illustrated in Fig. 7, where [47, 59] either fail to add any objects in the scene or place furnitures in unatural positions, thus resulting in bedrooms without beds (see 1st row Fig. 7) and scenes with overlapping furniture (see 2nd row Fig. 7). In contrary, our model successfullly generates plausible completions with multiple objects such as lamps, wardrobes and dressing tables.

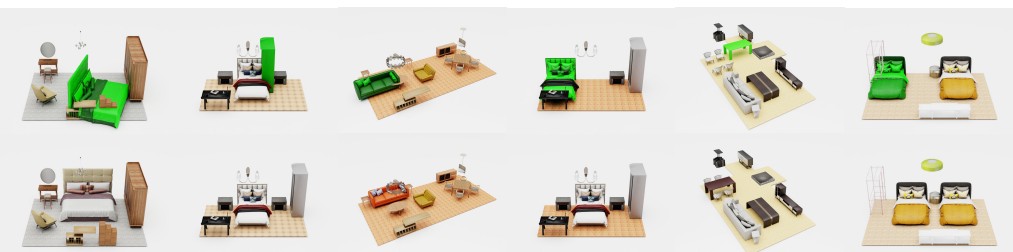

**Figure 8: Failure Case Detection and Correction**. We use a partial room with unnatural object arrangements. Our model identifies the problematic objects (first row, in green) and relocates them into meaningful positions.

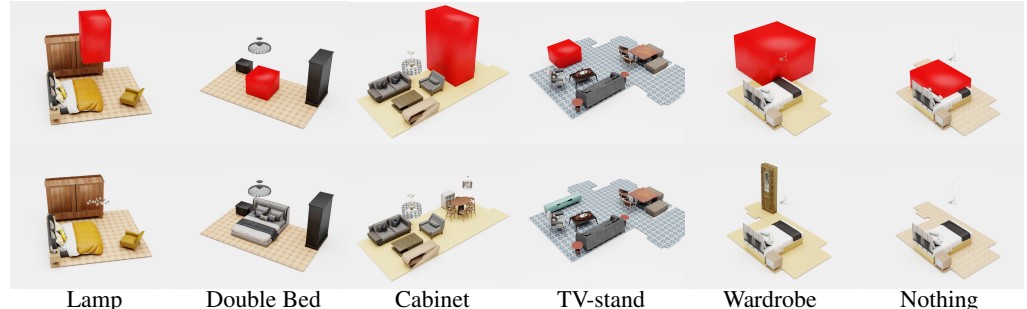

| Lamp | Double Bed | Cabinet | TV-stand | Wardrobe | Nothing |

**Figure 9: Object Suggestion**. A user specifies a region of acceptable positions to place an object (marked as red boxes, 1st row) and our model suggests suitable objects (2nd row) to be placed in this location.

**Failure Case Detection and Correction:** We showcase that our model is able to identify and correct unnatural object arrangements. Given a scene, we compute the likelihood of each object, according to our model, conditioned on the other objects in the scene. We identify problematic objects as those with low likelihood and sample a new location from our generative model to rearrange it. We test our model in various scenarios such as overlapping objects, objects outside the room boundaries and objects in unnatural positions and show that it successfully identifies problematic objects (highlighted in green in Fig. 8) and rearranges them into a more plausible position. Note that this task cannot be performed by methods that consider ordering because they assign very low likelihood to common objects appearing after rare objects e.g. beds after cabinets.

**Object Suggestion:** We now test the ability of our model to provide object suggestions given a scene and user specified location constraints. To perform this task we sample objects from our generative model and accept the ones that fullfill the constraints provided by the user. Fig. 9 shows examples of location constraints (red box in top row) and the corresponding objects suggested (bottom row). Note that even when the user provided region is partially outside the room boundaries (4th, 5th column), suggested objects always reside in the room. Moreover, if the acceptable region overlaps with another object, our model suggests adding nothing (6th column). This task requires computing the likelihood of an object conditioned on an arbitrary scene, which [59, 47] cannot perform due to ordering.

## 4.3 Perceptual Study

We conducted two paired Amazon Mechanical Turk perceptual studies to evaluate the quality of our generated layouts against [47] and [59]. We sample 6 bedroom layouts for each method from the same 211 test set floor plans. Users saw 2 different rotating 3D scenes per method randomly selected from 6 pre-rendered layouts. Random layouts for each floor plan were assessed by 5 different workers to evaluate agreement and diversity across samples for a total of 1055 question sets per paired study. Generated scenes of [47] were judged to contain errors like interpenetrating furniture 41.4% of the time, nearly twice as frequently as our method, while [59] performs significantly worse (Tab. 4). Regarding realism, the scenes of [47] were more realistic than ours in only 26.9% of the cases. We conclude that our method outperforms the baselines in the key metric, generation of realistic indoor scenes, by a large margin. Additional details are provided in the supplementary.

| Method | Condition | Mean Error Frequency ↓ | More ↑ Realistic | Realism CI 99% |
|---|---|---|---|---|
| FastSynth [47] | vs. Ours | 0.414 | 0.269 | [0.235, 0.306] |
| SceneFormer [59] | vs. Ours | 0.713 | 0.165 | [0.138, 0.196] |
| Ours | vs. Both | **0.232** | **0.783** | [0.759, 0.805] |

**Table 4: Perceptual Study Results**. Aggregated results for two A/B paired tests. Our method was judged more realistic with high confidence (binomial confidence interval with $\alpha = 0.01$ reported) and contained fewer errors.

## 5 Conclusion

We introduced ATISS, a novel autoregressive transformer architecture for synthesizing 3D rooms as unordered sets of objects. Our method generates realistic scenes that advance the state-of-the-art for scene synthesis. In addition, our novel formulation enables new interactive applications for semi-automated scene authoring, such as general scene completion, object suggestions, anomaly

detection and more. We believe that our model is an important step not only toward automating the generation of 3D environments, with impact on simulation and virtual testing, but also toward a new generation of tools for user-driven content generation. By accepting a wide range of user inputs, our model mitigates societal risks of task automation, and promises to usher in tools that enhance the workflow of skilled laborers, rather than replacing them. In future work, we plan to extend order invariance to object attributes to further expand interactive possibilities of this model, and to incorporate style information. As any machine learning model, our model can introduce learned biases for indoor scenes, and we plan to investigate learning from less structured and more widely available data sources to make this model applicable to a wider range of cultures and environments.

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
