# Supplementary Material for
# ATISS: Autoregressive Transformers for Indoor Scene Synthesis

**Despoina Paschalidou**[*,1,3,4]    **Amlan Kar**[4,5,6]    **Maria Shugrina**[4]    **Karsten Kreis**[4]

**Andreas Geiger**[1,2,3]    **Sanja Fidler**[4,5,6]

[1]Max Planck Institute for Intelligent Systems Tübingen    [2]University of Tübingen
[3]Max Planck ETH Center for Learning Systems
[4]NVIDIA    [5]University of Toronto    [6]Vector Institute
{firstname.lastname}@tue.mpg.de    {amlank, mshugrina, kkreis, sfidler}@nvidia.com

## Abstract

In this **supplementary document**, we provide a detailed overview of our network architecture and the training procedure. Subsequently, we describe the preprocessing steps that we followed to filter out problematic rooms from the 3D-FRONT dataset [7]. Next, we provide ablations on how different components of our system impact the performance of our model on the scene synthesis task and we compare ATISS with various transformer models that consider ordering. Finally, we provide additional qualitative and quantitative results as well as additional details for our perceptual study presented in Sec 4.3 in our main submission.

## 1    Implementation Details

In this section, we provide a detailed description of our network architecture. We then describe our training protocol and provide details on the metrics computation during training and testing. Finally, we also provide additional details regarding our baselines.

### 1.1    Network Architecture

Here we describe the architecture of each individual component of our model (from Fig. 2 in the main submission). Our architecture comprises four components: (i) the *layout encoder* that maps the room shape to a global feature representation $\mathbf{F}$, (ii) the *structure encoder* that maps the $M$ objects in a scene into per-object context embeddings $\mathbf{C} = \{\mathbf{C}_j\}_{j=1}^{M}$, (iii) the *transformer encoder* that takes $\mathbf{F}$, $\mathbf{C}$ and a query embedding $\mathbf{q}$ and predicts the features $\hat{\mathbf{q}}$ for the next object to be generated and (iv) the *attribute extractor* that autoregressively predicts the attributes of the next object.

**Layout Encoder:** The first part of our architecture is the *layout encoder* that is used to map the room's floor into a global feature representation $\mathbf{F}$. We follow [30] and we model the floor plan with its top-down orthographic projection. This projection maps the floor plan into an image, where pixel values of $1$ indicate regions inside the room and pixel values of $0$ otherwise. The layout encoder is implemented with a ResNet-18 architecture [10] that is not pre-trained on ImageNet [2]. We empirically observed that using a pre-trained ResNet resulted in worse performance. From the

---

[*]Work done during Despoina's internship at NVIDIA.

35th Conference on Neural Information Processing Systems (NeurIPS 2021).

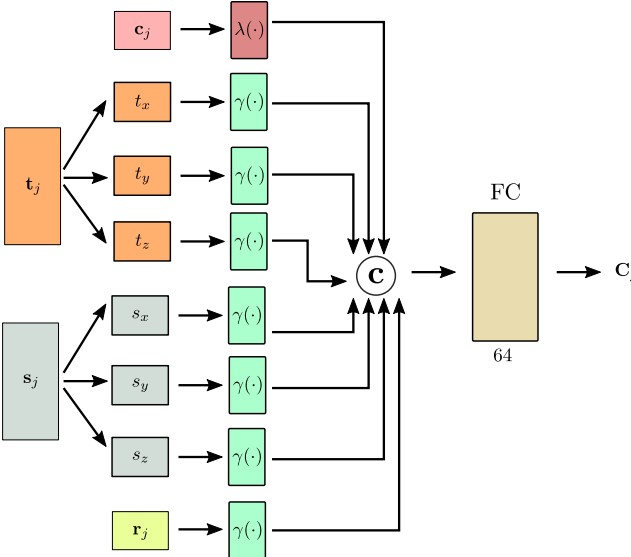

Figure 1: **Structure Encoder.** The structure encoder predicts the per-object context embeddings $\mathbf{C}_j$ conditioned on the object attributes. For the object category $\mathbf{c}_j$, we use a learnable embedding $\lambda(\cdot)$, whereas for the location $\mathbf{t}_j$, the size $\mathbf{s}_j$ and orientation $\mathbf{r}_j$ we employ the positional encoding from (1). Note that the positional encoding $\gamma(\cdot)$ is applied separately in each dimension of $\mathbf{t}_j$ and $\mathbf{s}_j$.

original architecture, we remove the final fully connected layer and replace it with a linear projection to $64$ dimensions, after average pooling.

**Structure Encoder:** The structure encoder maps the attributes of each object into a per-object context embedding $\mathbf{C}_j$. For the object category $\mathbf{c}_j$, we use a learnable embedding, which is simply a matrix of size $C \times 64$, that stores a per-object category vector, for all $C$ object categories in the dataset. For the size $\mathbf{s}_j$, the position $\mathbf{t}_j$ and the orientation $\mathbf{r}_j$, we use the positional encoding of [29] as follows

$$\gamma(p) = (\sin(2^0\pi p), \cos(2^0\pi p), \dots, \sin(2^{L-1}\pi p), \cos(2^{L-1}\pi p)) \tag{1}$$

where $p$ can be any of the size, position or orientation attributes and $\gamma(\cdot)$ is applied separately in each attribute's dimension. In our experiments, $L$ is set to 32. The output of each embedding layer, used to map the category, size, location and orientation in a higher dimensional space, are concatenated into an $512$-dimensional feature vector, which is then mapped to the per-object context embedding. A pictorial representation of the structure encoder is provided in Fig. 1.

**Transformer Encoder:** We follow [29, 3] and implement our transformer encoder as a multi-head attention transformer without any positional encoding. Our transformer consists of $4$ layers with $8$ attention heads. The queries, keys and values have $64$ dimensions and the intermediate representations for the MLPs have 1024 dimensions. To implement the transformer architecture we use the transformer library provided by Katharopoulos et al. [12][2]. The input set of the transformer is $\mathbf{I} = \{\mathbf{F}\} \cup \{\mathbf{C}_j\}_{j=1}^M \cup \mathbf{q}$, where $M$ denotes the number of objects in the scene and $\mathbf{q} \in \mathbb{R}^{64}$ is a learnable object query vector that allows the transformer to predict output features $\hat{\mathbf{q}} \in \mathbb{R}^{64}$ used for generating the next object to be added in the scene.

**Attribute Extractor:** The attribute extractor autoregressively predicts the attributes of the next object to be added in the scene. The MLP for the object category is a linear layer with $64$ hidden dimensions that predicts $C$ class probabilities per object. The MLPs for the location, orientation and size predict the mean, variance and mixing coefficient for the $K$ logistic distributions for each attribute. In our experiments we set $K = 10$. The size, location and orientation attributes are predicted using a 2-layer MLP with RELU non-linearities with hidden size 128 and output size 64. A pictorial representation for the MLPs $t_\theta(\cdot)$ and $\sigma_\theta(\cdot)$ used to predict the parameters of the mixture of logistics distribution for the location and the size is provided in Fig. 2. Note that $r_\theta$ is defined in a similar manner.

[2]https://github.com/idiap/fast-transformers

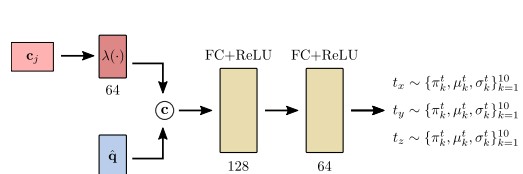 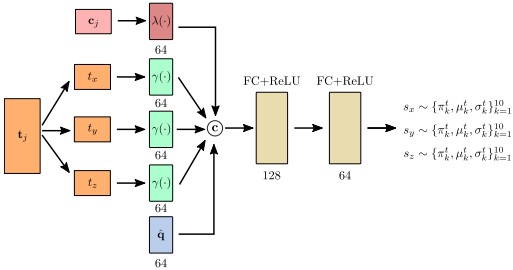

(a) $t_\theta(\cdot)$ predicts the parameters of the mixture of logistics distribution for the location $\mathbf{t}$.

(b) $s_\theta(\cdot)$ predicts the parameters of the mixture of logistics distribution for the size $\mathbf{s}$.

Figure 2: **Attribute Extractor.** The attribute extractor consists of four MLPs that autoregressively predict the object attributes. Here we visualize the MLP $t_\theta(\cdot)$ for the location attribute (left side) and the MLP $s_\theta(\cdot)$ for the size attribute (right side).

## 1.2 Object Retrieval

During inference, we select 3D models from the 3D-FUTURE dataset [8] to be placed in the scene based on the predicted category, location, orientation and size. In particular, we perform nearest neighbor search through the 3D-FUTURE dataset[8] to find the closest model in terms of object dimensions. While prior work [24, 31] explored more complex object retrieval schemes based on object dimensions and object cooccurrences (i.e. favor 3D model of objects that frequently co-occur in the dataset), we note that our simple object retrieval strategy consistently resulted in visually plausible rooms. We leave more advanced object retrieval schemes for future research.

## 1.3 Training Protocol

In all our experiments, we use the Adam optimizer [13] with learning rate $\eta = 10^{-4}$ and no weight decay. For the other hyperparameters of Adam we use the PyTorch defaults: $\beta_1 = 0.9$, $\beta_2 = 0.999$ and $\epsilon = 10^{-8}$. We train all models with a batch size of 128 for 100k iterations. During training, we perform rotation augmentation with random rotations between $[0, 360]$ degrees. To determine when to stop training, we follow common practice and evaluate the validation metric every 1000 iterations and use the model that performed best as our final model.

## 1.4 Metrics Computation

As mentioned in our main submission, we evaluate our model and our baselines using the KL divergence between the object category distributions of synthesized and real scenes and the classification accuracy of a classifier trained to discriminate real from synthetic scenes as well as the FID [11] score between $256^2$ top-down orthographic projections of synthesized and real scenes using the code provided by Parmar et al. [19][3]. For the metrics computation, we generate the same amount of scenes as in the test set and we compute each metric using real scenes from the test set. In particular, for the KL divergence, we measure the frequency of object category occurrences in the generated scenes and compare it with the frequency of object occurrences in real scenes. Regarding the scene classification accuracy, we train a classifier to distinguish real from generated scenes. Our classifier is an Alexnet [14] pre-trained on ImageNet, that takes as input a $256^2$ top-down image-based representation of a room and predicts whether this scene is real or synthetic. Both for the FID and the classification accuracy, we repeat the metric computation 10 times and report the average.

## 1.5 Baselines

In this section, we provide additional details regarding our baselines. We compare our model with FastSynth [24] and SceneFormer [31]. Both methods were originally evaluated on the SUNCG dataset [26], which is currently unavailable, thus we retrained both on 3D-FRONT using the augmentation techniques described in the original papers. To ensure fair comparison, we use the same object retrieval for all methods and no rule-based post-processing on the generated layouts.

---

[3]https://github.com/GaParmar/clean-fid

**FastSynth:** In FastSynth [24], the authors employ a series of image-based CNNs to sequentially predict the attributes of the next object to be added in the scene. In addition to 2D labeled bounding boxes they have auxiliary supervision in the form of object segmentation masks, depth maps, wall masks etc. For more details, we refer the reader to [30]. During training, they assume that there exists an ordering of objects in each scene, based on the average size of each category multiplied by its frequency of occurrences in the dataset. Each CNN module is trained separately and the object properties are predicted in an autoregressive manner: object category first, followed by location, orientation and size. We train [24][4] using the provided PyTorch [23] implementation with the default parameters until convergence.

**SceneFormer:** In SceneFormer [31], the authors utilize a series of transformers to autoregressively add objects in the scene, similar to [24]. In particular, they train a separate transformer for each attribute and they predict the object properties in an autoregressive manner: object category first, followed by orientation, location and size. Similar to [24], they also treat scenes as ordered sequences of objects ordered by the frequency of their categories. We train [31][5] using the provided PyTorch [23] implementation with the default parameters until convergence.

## 2  3D-FRONT Dataset Filtering

We evaluate our model on the 3D-FRONT dataset [7], which is one of the few available datasets that contain indoor environments. 3D-FRONT contains a collection of 6813 houses with roughly 14629 designed rooms, populated with 3D furniture objects from the 3D-FUTURE dataset [8]. In our experiments, we focused on four room types: (i) bedrooms, (ii) living rooms, (iii) dining rooms and (iv) libraries. Unfortunately, 3D-FRONT contains multiple problematic rooms with unnatural sizes, misclassified objects as well as objects in unnatural positions e.g. outside the room boundaries, lamps on the floor, overlapping objects etc. Therefore, in order to be able to use it, we had to perform thorough filtering to remove problematic scenes. In this section, we present in detail the pre-processing steps for each room type. We plan to release the names/ids of the filtered rooms, when the paper is published.

The 3D-FRONT dataset provides scenes for the following room types: *bedroom, diningroom, elderly-room, kidsroom, library, livingdiningroom, livingroom, masterbedroom, nannyroom, secondbedroom* that contain 2287, 3233, 233, 951, 967, 2672, 1095, 3313, 16 and 2534 rooms respectively. Since some room types have very few rooms we do not consider them in our evaluation.

**Bedroom:** To create training and test data for bedroom scenes, we consider rooms of type *bedroom, secondbedroom* and *masterbedroom*, which amounts to 8134 rooms in total. We start by removing rooms of unnatural sizes, namely rooms that are larger than 6m × 6m in floor size and taller than 4m. Next, we remove infrequent objects that appear in less than 15 rooms, such as chaise lounge sofa, l-shaped sofa, barstool, wine cabinet etc. Subsequently, we filter out rooms that contain fewer than 3 and more than 13 objects, since they amount to a small portion of the dataset. Since the original dataset contained various rooms with problematic object arrangements such overlapping objects, we also remove rooms that have objects that are overlapping as well as misclassified objects e.g. beds being classified as wardrobes. This results in 5996 bedrooms with 21 object categories in total. Fig. 3a illustrates the number of appearances of each object category in the 5996 bedroom scenes and we remark that the most common category is the nightstand with 8337 occurrences and the least common is the coffee table with 45.

**Library:** We consider rooms of type *library* that amounts to 967 scenes in total. For the case of libraries, we start by filtering out rooms with unnatural sizes that are larger than 6m × 6m in floor size and taller than 4m. Again we remove rooms that contain overlapping objects, objects positioned outside the room boundaries as well as rooms with unnatural layouts e.g. single chair positioned in the center of the room. We also filter out rooms that contain less than 3 objects and more than 12 objects since they appear less frequently. Our pre-processing resulted in 622 rooms with 19 object categories in total. Fig. 3b shows the number of appearances of each object category in the 622

---

[4]https://github.com/brownvc/fast-synth
[5]https://github.com/cy94/sceneformer

libraries. The most common category is the bookshelf with 1109 occurrences and the least common is the wine cabinet with 19.

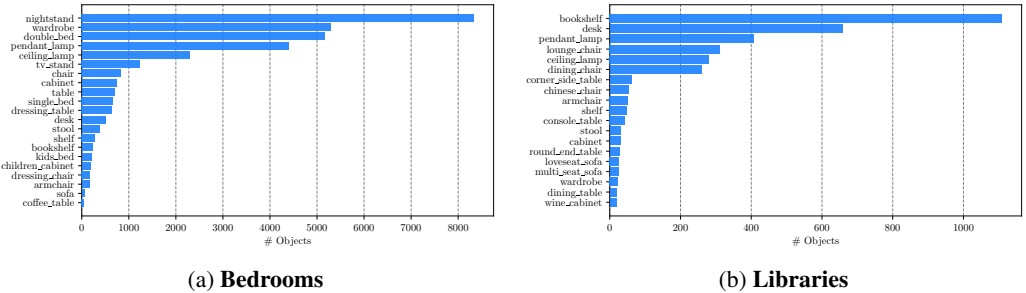

(a) **Bedrooms**        (b) **Libraries**

Figure 3: **Number of object occurrences in Bedrooms and Libraries.**

**Living Room:** For the living rooms, we consider rooms of type *livingroom* and *livingdiningroom*, which amounts to 3767 rooms. We follow a similar process as before and we start by filtering out rooms with unnatural sizes. In particular, we discard rooms that are larger than $12m \times 12m$ in floor size and taller than $4m$. We also remove uncommon objects that appear in less than 15 rooms such as bed and bed frame. Next, we filter out rooms that contain less than 3 objects and more than 13 objects, since they are significantly less frequent. For the case of living rooms, we observed that some of the original scenes contained multiple lamps without having any other furniture. Since this is unnatural, we also removed these scenes together with some rooms that had either overlapping objects or objects positioned outside the room boundaries. Finally, we also remove any scenes that contain any kind of bed e.g. double bed, single bed, kid bed etc. After our pre-processing, we ended up with 2962 living rooms with 24 object categories in total. Fig. 4a visualizes the number of occurrences of each object category in the living rooms. We observe that the most common category is the dining chair with 9009 occurrences and the least common is the chaise lounge sofa with 30.

**Dining Room:** For the dining rooms, we consider rooms of type *diningroom* and *livingdiningroom*, since the *diningroom* scenes amount to only 233 scenes. This results in 3233 rooms in total. For the dining rooms, we follow the same filtering process as for the living rooms and we keep 2625 rooms with 24 objects in total. Fig. 4b shows the number of occurrences of each object category in the dining rooms. The most common category is the dining chair with 9589 occurrences and the least common is the chaise lounge sofa with 19.

To generate the train, test and validation splits, we split the preprocessed rooms such that 70% is used for training, 20% for testing and 10% for validation. Note that the 3D-FRONT dataset comprises multiple houses that may contain the same room, e.g the exact same object arrangement might appear in multiple houses. Thus splitting train and test scenes solely based on whether they belong to different houses could result in the same room appearing both in train and test scenes. Therefore, instea of randomly selecting rooms from houses but we select from the set of rooms with distinct object arrangements.

## 3 Ablation Study

In this section, we investigate how various components of our model affect its performance on the scene synthesis task. In Sec. 3.1, we investigate the impact of the number of logistic distributions in the performance of our model. Next, in Sec. 3.2, we examine the impact of the architecture of the layout encoder. In Sec. 3.3, we compare ATISS with two variants of our model that consider ordered sets of objects. Unless stated otherwise, all ablations are conducted on the bedroom scenes of the 3D-FRONT [7] dataset.

### 3.1 Mixture of Logistic distributions

We represent objects in a scene as labeled 3D bounding boxes and model them with four random variables that describe their category, size, orientation and location, $o_j = \{\mathbf{c}_j, \mathbf{s}_j, \mathbf{t}_j, \mathbf{r}_j\}$. The category $\mathbf{c}_j$ is modeled using a categorical variable over the total number of object categories $C$ in the

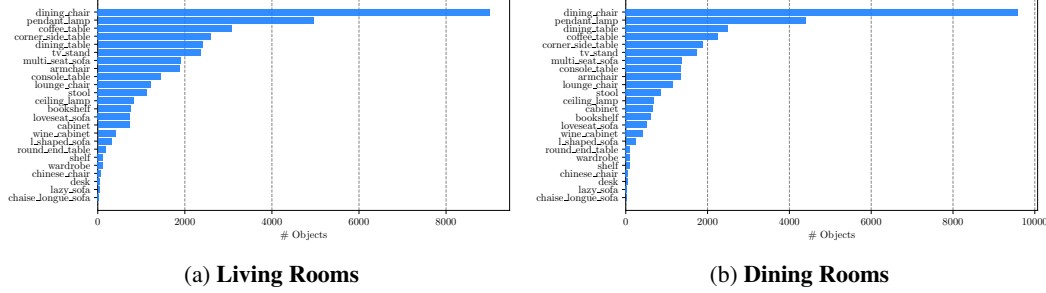

(a) **Living Rooms**  (b) **Dining Rooms**

Figure 4: **Number of object occurrences in Living Rooms and Dining Rooms.**

dataset. For the size $\mathbf{s}_j \in \mathbb{R}^3$, the location $\mathbf{t}_j \in \mathbb{R}^3$ and the orientation $\mathbf{r}_j \in \mathbb{R}^1$, we follow [25, 28] and model them with a mixture of logistic distributions

$$\mathbf{s}_j \sim \sum_{k=1}^{K} \pi_k^s \mathrm{logistic}(\mu_k^s, \sigma_k^s) \quad \mathbf{t}_j \sim \sum_{k=1}^{K} \pi_k^t \mathrm{logistic}(\mu_k^t, \sigma_k^t) \quad \mathbf{r}_j \sim \sum_{k=1}^{K} \pi_k^r \mathrm{logistic}(\mu_k^r, \sigma_k^r) \quad (2)$$

where $\pi_k^s$, $\mu_k^s$ and $\sigma_k^s$ denote the weight, mean and variance of the $k$-th logistic distribution used for modeling the size. Similarly, $\pi_k^t$, $\mu_k^t$ and $\sigma_k^t$ and $\pi_k^r$, $\mu_k^r$ ans $\sigma_k^r$ refer to the weight, mean and variance of the $k$-th logistic of the location and orientation, respectively.

In this experiment, we test our model with different numbers for logistic distributions for modelling the object attributes. Results are summarized in Tab. 1.

|  | FID ($\downarrow$) | Classification Accuracy ($\downarrow$) | Category Distribution ($\downarrow$) |
|---|---|---|---|
| $K = 1$ | $41.71 \pm 0.4008$ | $0.7826 \pm 0.0080$ | $0.0491$ |
| $K = 5$ | $40.41 \pm 0.2491$ | $0.5667 \pm 0.0405$ | $0.0105$ |
| $K = 10$ | $\mathbf{38.39} \pm 0.3392$ | $\mathbf{0.5620} \pm 0.0228$ | $0.0085$ |
| $K = 15$ | $40.41 \pm 0.4504$ | $0.5980 \pm 0.0074$ | $0.0095$ |
| $K = 20$ | $40.39 \pm 0.3964$ | $0.6680 \pm 0.0035$ | $\mathbf{0.0076}$ |

Table 1: **Ablation Study on the Number of Logistic Distributions.** This table shows a quantitative comparison of our approach with different numbers of $K$ logistic distributions for modelling the size, the location and the orientation of each object.

As it is expected, using a single logistic distribution (first row in Tab. 1) results in worse performance, since it does not have enough representation capacity for modelling the object attributes. We also note that increasing the number of logistic distributions beyond 10 hurts performance wrt. FID and classification accuracy. We hypothesize that this is due to overfitting. In our experiments we set $K = 10$.

## 3.2 Layout Encoder

We further examine the impact of the layout encoder on the performance of our model. To this end, we replace the ResNet-18 architecture [10], with an AlexNet [14]. From the original architecture, we remove the final classifier layers and keep only the feature vector of length 9216 after max pooling. We project this feature vector to 64 dimensions with a linear projection layer. Similar to our vanilla model, we do not use an AlexNet pre-trained on ImageNet because we empirically observed that it resulted in worse performance.

|  | FID ($\downarrow$) | Classification Accuracy ($\downarrow$) | Category Distribution ($\downarrow$) |
|---|---|---|---|
| AlexNet | $40.40 \pm 0.2637$ | $0.6083 \pm 0.0034$ | $\mathbf{0.0064}$ |
| ResNet-18 | $\mathbf{38.39} \pm 0.3392$ | $\mathbf{0.5620} \pm 0.0228$ | $0.0085$ |

Table 2: **Ablation Study on the Layout Encoder Architecture.** This table shows a quantitative comparison of ATISS with two different layout encoders.

Tab. 2 compares the two variants of our model wrt. to the FID score, the classification accuracy and the KL-divergence. We remark that our method is not particularly sensitive to the choice of the

| | FID (↓) | Classification Accuracy (↓) | Category Distribution (↓) |
|---|---|---|---|
| Ours+Perm+Order | 40.18 ± 0.2831 | 0.6019 ± 0.0060 | 0.0089 |
| Ours+Order | 38.67 ± 0.5552 | 0.7603 ± 0.0010 | 0.0533 |
| Ours | **38.39** ± 0.3392 | **0.5620** ± 0.0228 | 0.0085 |

Table 3: **Ablation Study on Ordering.** This table shows a quantitative comparison of our approach wrt. two variants of our model that represent rooms as ordered sequence of objects.

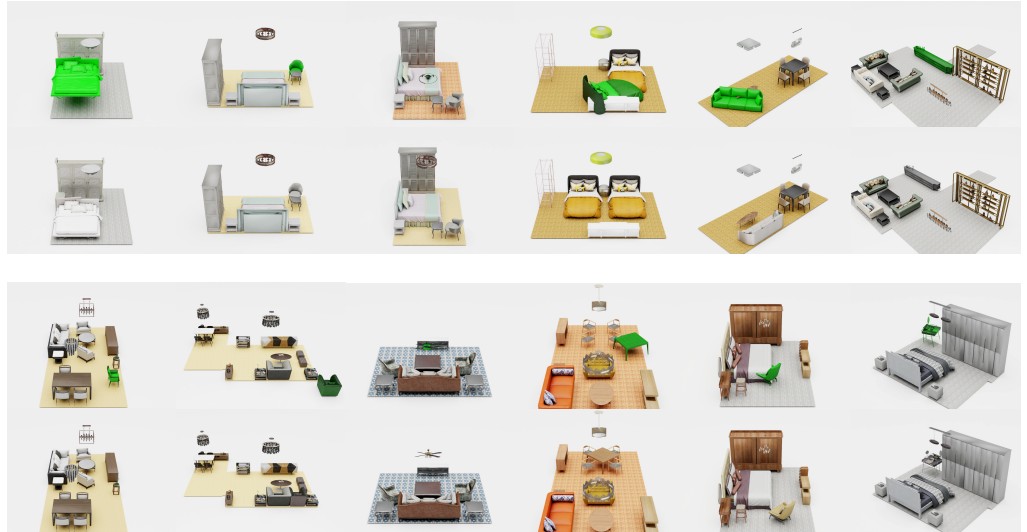

Figure 5: **Failure Case Detection and Correction**. Starting from a room with an unnatural object arrangement, our model identifies the problematic objects (first row and third row, in green) and relocates them into meaningful positions (second and fourth row).

layout encoder. However, using an AlexNet results in slightly worse performance, hence we utilize a ResNet-18 in all our experiments.

## 3.3 Transformers with Ordering

In this section, we analyse the benefits of synthesizing rooms as unordered sets of objects in contrast to ordered sequences. To this end, we train two variants of our model that utilize a positional embedding [29] to incorporate order information to the input. The first variant is trained with random permutations of the input (Ours+Perm+Order), similar to our model, whereas the second with a fixed ordering based on the object frequency (Ours+Order) as described in [24, 31]. We compare these variants to our model on the scene synthesis task and observe that the variant with the fixed ordering (second row Tab. 3) performs significantly worse as the classifier can identify synthesized scenes with 76% accuracy. Moreover, we remark that besides enabling all the applications presented in our main submission, training with random permutations also improves the quality of the synthesized scenes (first row Tab. 3). However, our model that is permutation invariant, namely the prediction is the same regardless of the order of the partial scene, performs even better (third row Tab. 3). We conjecture that the invariance of our model will be more even more crucial for training with either larger datasets or larger scenes i.e. scenes with more objects, because observing a single order allows reasoning about all permutations of the partial scene.

## 4 Applications

In this section, we provide additional qualitative results for various interactive applications that benefit greatly by our unordered set formulation.

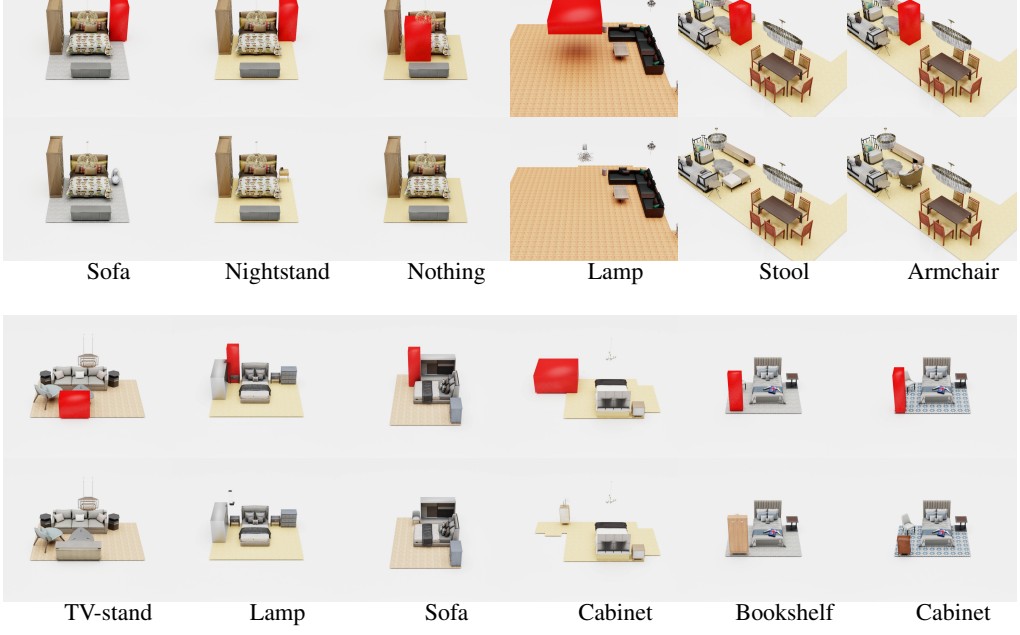

| | | | | | |
|---|---|---|---|---|---|
| Sofa | Nightstand | Nothing | Lamp | Stool | Armchair |

| | | | | | |
|---|---|---|---|---|---|
| TV-stand | Lamp | Sofa | Cabinet | Bookshelf | Cabinet |

Figure 6: **Object Suggestion**. A user specifies a region of acceptable positions to place an object (marked as red boxes, first and third row) and our model suggests suitable objects (second and fourth row) to be placed in this location.

## 4.1 Failure Case Detection And Correction

In this experiment, we investigate whether our model is able to identify unnatural furniture layouts and reposition the problematic objects such that they preserve their functional properties. As we described in our main submission, we identify problematic objects as those with low likelihood and as soon as a problematic object is identified, we sample a new location from our generative model to reposition it. Fig. 5 shows additional qualitative results on this task. The first and third row show examples of unnatural object arrangements, together with the problematic object, highlighted in green, for each scenario. We note that our model successfully identifies objects in unnatural positions e.g. flying bed (first row, first column Fig. 5), light inside the bed (first row, third column Fig. 5) or table outside the room boundaries (third row, fourth column Fig. 5 ) as well as problematic objects that do not necessarily look unnatural, such as a cabinet blocking the corridor (first row, sixth column Fig. 5), a chair facing the wall (third row, first column Fig. 5) or a lamp being too close to the table (third row, third column Fig. 5). After having identified the problematic object, our model consistently repositions it at plausible position.

## 4.2 Object Suggestion

For this task, we examine the ability of our model to provide object suggestions given a scene and user specified location constraints. For this experiment, the user only provides location constraints, namely valid positions for the centroid of the object to be generated. Fig. 6 shows examples of the location constraints, marked with red boxes, (first and third row) and the corresponding objects suggested by our model (second and fourth row). We observe that our model consistently makes plausible suggestions, and for the cases that a user specifies a region that overlaps with other objects in the scene, our model suggests adding nothing (first row, third column Fig. 6). In Fig. 6, we also provide two examples, where our model makes different suggestions based on the same location constraints, such as sofa and nightstand for the scenario illustrated in the first and second column and stool and armchair for the scenario illustrated in the fifth and sixth column in the first row.

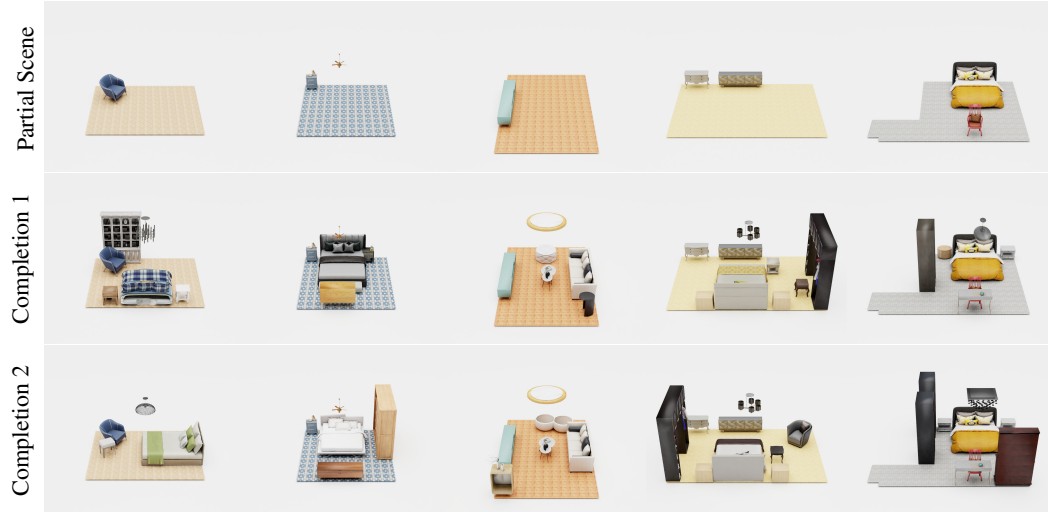

Figure 7: **Scene Completion**. Starting from a partially complete scene (first row), we visualize two examples of scene completions using our model (second and third row).

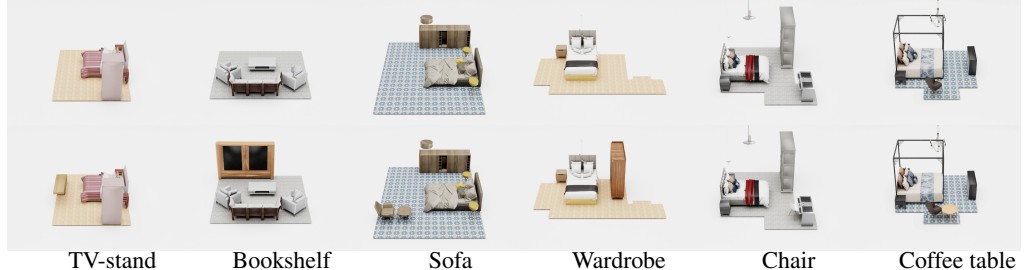

| TV-stand | Bookshelf | Sofa | Wardrobe | Chair | Coffee table |

Figure 8: **Object Placement**. Starting from a partially complete scene, the user specifies an object to be added in the scene and our model places it at a reasonable position. The first rows illustrates the starting scene and the second row the generated scened using the user specified object (third row).

## 4.3   Scene Completion

Starting from a partial scene, we want to evaluate the ability of our model to generate plausible object arrangements. To generate the partial scenes, we randomly sample scenes from the test set and remove the majority of the objects in them. Fig. 7 shows examples for various partial rooms (first row Fig. 7), as well as two alternative scene completions using our model (second and third row Fig. 7). We observe that our model generates diverse arrangements of objects that are consistently meaningful. For example, for the case where the partial scene consists of a chair and a bed (last column Fig. 7), our model generates completions that have nightstands surrounding the bed as well as a desk in front of the chair.

## 4.4   Object Placement

Finally, we showcase the ability of our model to add a specific object in a scene on demand. Fig. 8 illustrates the original scene (first row) and the complete scene (second row) using the user specified object (third row). To perform this task, we condition on the given scene and instead of sampling from the predicted object category distribution, we use the user provided object category and sample the rest of the object attributes i.e. translation, size and orientation. Also in this task, we note that the generated objects are realistic and match the room layout.

# 5 Scene Synthesis

In this section, we provide additional qualitative results for our scene synthesis experiment on the four 3D-FRONT rooms. Moreover, since, we repeat the FID score and classification accuracy computation 10 times, in Tab. 4, we also report the standard deviation for completeness.

| | FID Score (↓) | | | Scene Classification Accuracy | | | Category KL Divergence (↓) | | |
|---|---|---|---|---|---|---|---|---|---|
| | FastSynth | SceneFormer | Ours | FastSynth | SceneFormer | Ours | FastSynth | SceneFormer | Ours |
| Bedrooms | $40.89 \pm 0.5098$ | $43.17 \pm 0.6921$ | $\mathbf{38.39} \pm 0.3392$ | $0.883 \pm 0.0010$ | $0.945 \pm 0.0009$ | $\mathbf{0.562} \pm 0.0228$ | 0.0064 | **0.0052** | 0.0085 |
| Living | $61.67 \pm 1.2136$ | $69.54 \pm 0.9542$ | $\mathbf{33.14} \pm 0.4204$ | $0.945 \pm 0.0010$ | $0.972 \pm 0.0010$ | $\mathbf{0.516} \pm 0.0075$ | 0.0176 | 0.0313 | **0.0034** |
| Dining | $55.83 \pm 1.0078$ | $67.04 \pm 1.3043$ | $\mathbf{29.23} \pm 0.3533$ | $0.935 \pm 0.0019$ | $0.941 \pm 0.0008$ | $\mathbf{0.477} \pm 0.0027$ | 0.0518 | 0.0368 | **0.0061** |
| Library | $37.72 \pm 0.4501$ | $55.34 \pm 0.1056$ | $\mathbf{35.24} \pm 0.2683$ | $0.815 \pm 0.0032$ | $0.880 \pm 0.0009$ | $\mathbf{0.521} \pm 0.0048$ | 0.0431 | 0.0232 | **0.0098** |

Table 4: **Quantitative Comparison.** We report the FID score (↓) at $256^2$ pixels, the KL divergence (↓) between the distribution of object categories of synthesized and real scenes and the real vs. synthetic classification accuracy for all methods. Classification accuracy closer to 0.5 is better.

Conditioned on a floor plan, we evaluate the performance of our model on generating plausible furniture arrangements and compare with FastSynth [24] and SceneFormer [31]. Fig. 20 provides a qualitative comparison of generated bedroom scenes conditioned on the same floor layout using our model and our baselines. We observe that in contrast to [24, 31], our model consistently generates layouts with more diverse objects. In particular, [31] typically generates bedrooms that consist only of a bed, a wardrobe and less frequently also a nightstand, whereas both our model and FastSynth synthesize rooms with more diverse objects. Similarly generated scenes for living rooms and dining rooms are provided in Fig. 21 and Fig. 22 respectively. We observe that for the case of living rooms and dining rooms both baselines struggle to generate plausible object arrangements, namely generated objects are positioned outside the room boundaries, have unnatural sizes or populate a small part of the scene. We hypothesize that this might be related to the significantly smaller amount of training data compared to bedrooms. Instead our model, generates realistic living rooms and dining rooms. For the case of libraries (see Fig. 23), again both [24, 31] struggle to generate functional rooms.

## 5.1 Object Co-occurrence

To further validate the ability of our model to reproduce the probabilities of object co-occurrence in the real scenes, we compare the probabilities of object co-occurrence of synthesized scenes using our model, FastSynth [24] and SceneFormer [31] for all room types. In particular, in this experiment, we generate 5000 scenes using each method and report the difference between the probabilities of object co-occurrences between real and synthesized scenes. Fig. 9 summarizes the absolute differences for the bedroom scenes. We observe that our model better captures the object co-occurrence than baselines since the absolute differences for most object pairs are consistently smaller.

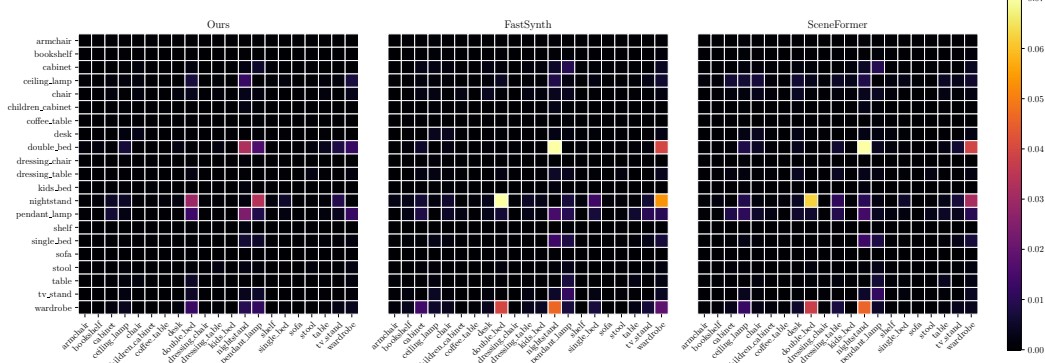

Figure 9: **Absolute Difference between Object Co-occurrence in Bedrooms.** We visualize the absolute difference of the probabilities of object co-occurrence computed between real and synthesized scenes using ATISS (left), FastSynth (middle) and SceneFormer (right). Larger differences correspond to warmer colors and are worse.

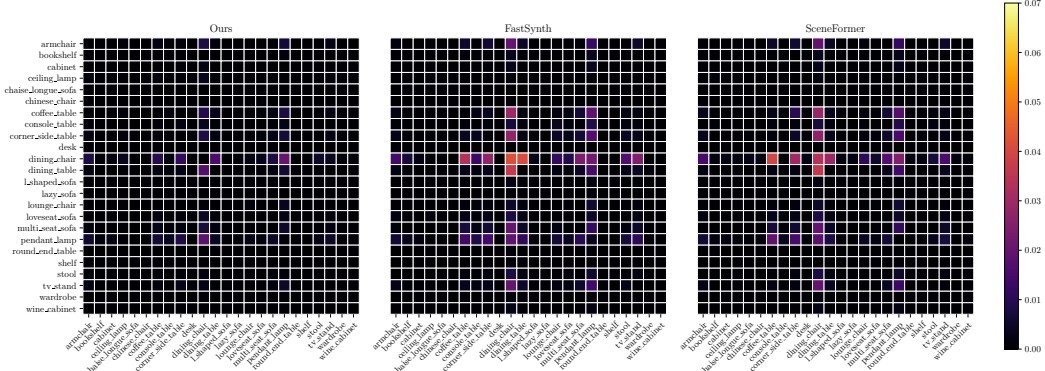

Figure 10: **Absolute Difference between Object Co-occurrence in Living Rooms.** We visualize the absolute difference of the probabilities of object co-occurrence computed between real and synthesized scenes using ATISS (left-most column), FastSynth (middle column), SceneFormer (right-most column). Lower is better.

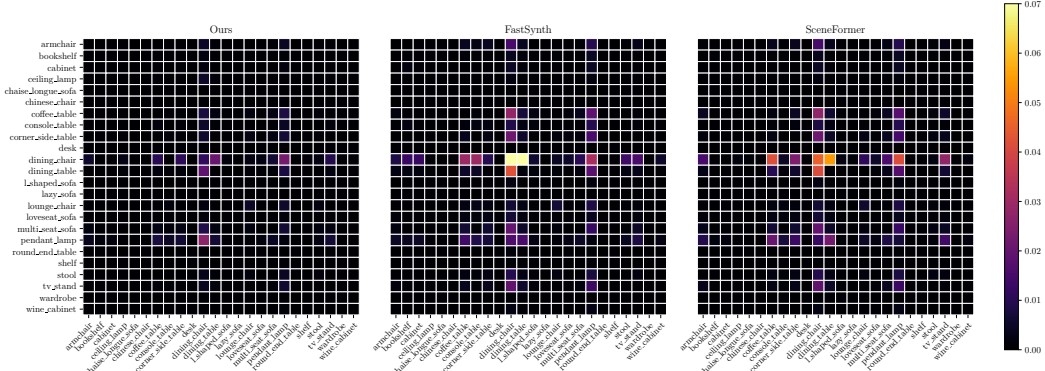

Figure 11: **Absolute Difference between Object Co-occurrence in Dining Rooms.** We visualize the absolute difference of the probabilities of object co-occurrence computed between real and synthesized scenes using ATISS (left-most column), FastSynth (middle column), SceneFormer (right-most column). Lower is better.

This is also validated for the case of living rooms (Fig. 10), dining rooms (Fig. 11) and libraries (Fig. 12), where our model better captures the object co-occurrences than both FastSynth [24] and SceneFormer [31]. Note that from our analysis it becomes evident that while our method better reproduces the probabilities of object co-occurrence from the real scenes, all methods are able to generate scenes with plausible object co-occurrences. This is expected, since learning the categories of objects to be added in a scene is a significantly easier task in comparison to learning their sizes and positions in 3D space.

Finally, in Fig. 13, we visualize the per-object difference in frequency of occurrence between synthesized and real scenes from the test set for all room types. We observe that our model generates object arrangements with comparable per-object frequencies to real rooms. In particular, for the case of living rooms (13b), dining rooms (13c) and libraries (13d) that are more challenging rooms types due to their smaller size, our model has an even smaller discrepancy wrt. the per-object frequencies.

## 5.2 Visualizations of Predicted Distributions

In this section, we provide examples of the predicted location distributions for different input scenes. In particular, we randomly select 6 bedroom floor plans and conditioned on them we generate 5000

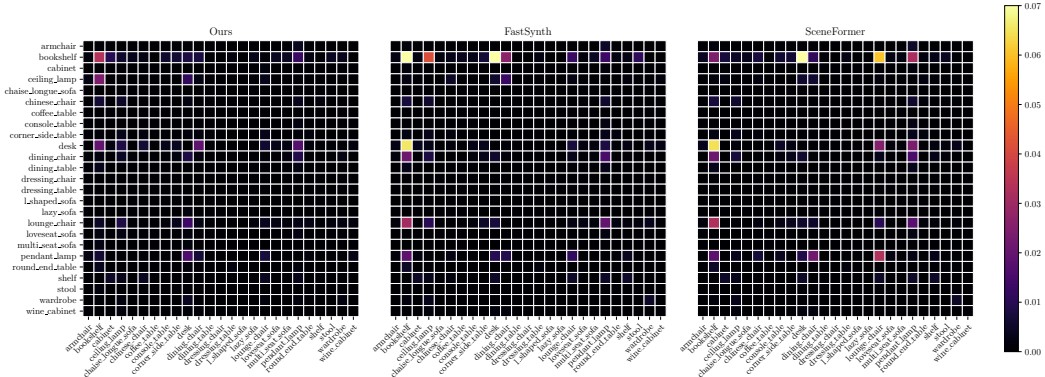

Figure 12: **Absolute Difference between Object Co-occurrence in Libraries.** We visualize the absolute difference of the probabilities of object co-occurrence computed between real and synthesized scenes using ATISS (left-most column), FastSynth (middle column), SceneFormer (right-most column). Lower is better.

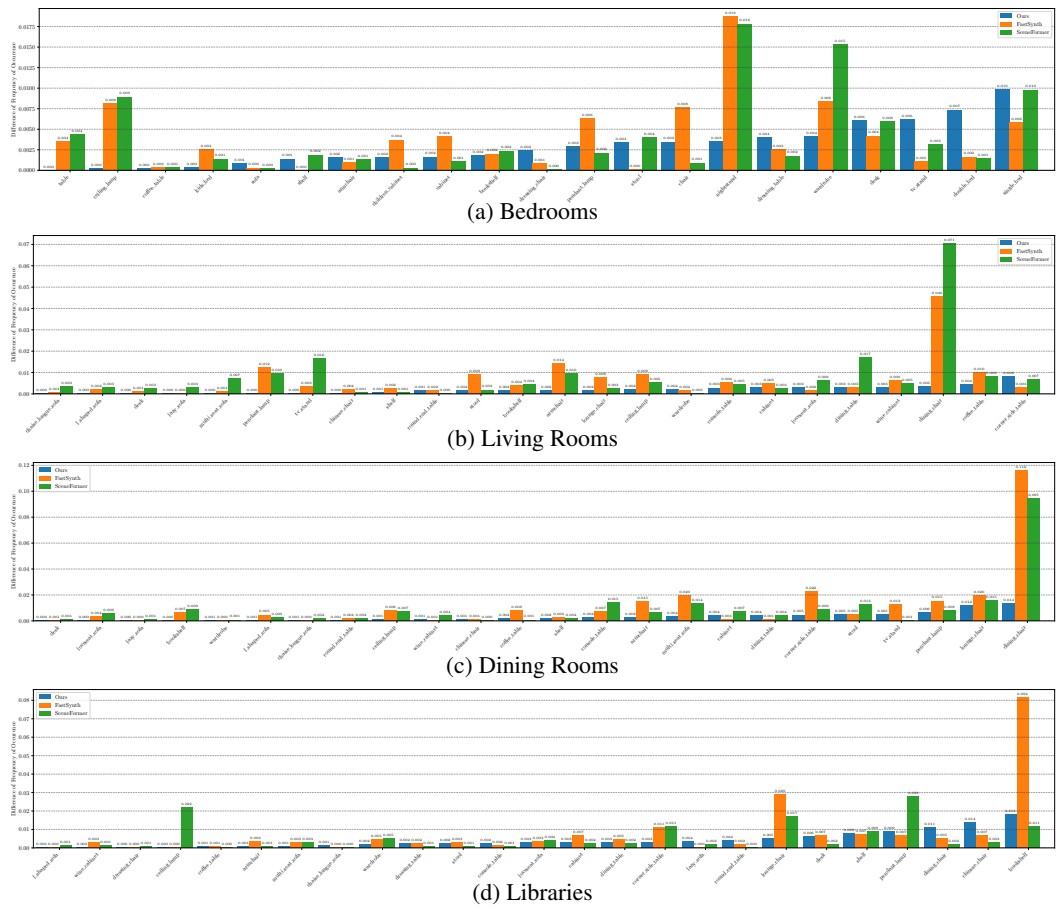

(a) Bedrooms

(b) Living Rooms

(c) Dining Rooms

(d) Libraries

Figure 13: **Difference of Per-Object Frequencies.** We visualize the absolute difference between the per-object frequency of generated and real scenes using our method, FastSynth [24] and Scene-Former [31] for all room types. Lower is better.

# Probability distributions for Chair

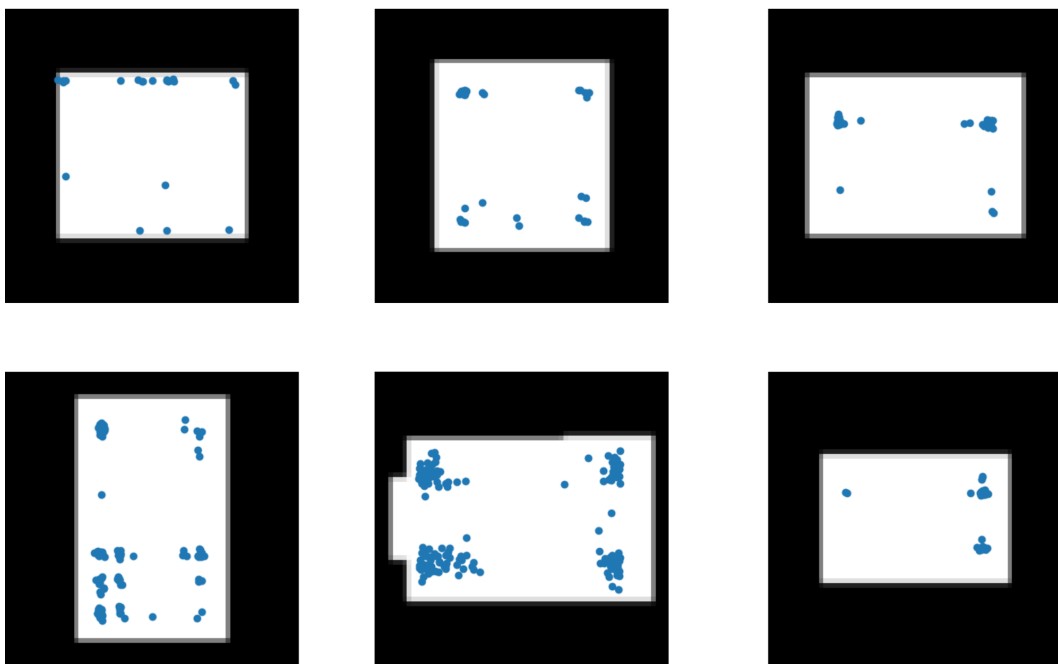

Figure 14: **Location Distributions for Chair.**

scenes conditioned on each floor plan. Based on the locations of the generated objects, we create scatter plots for the locations of various object categories i.e. chair (Fig. 14), desk (Fig. 15), nightstand (Fig. 16), wardrobe (Fig. 17). We observe that for all object categories the location distributions of the generated objects are consistently meaningful.

## 5.3 Computational Requirements

In this section, we provide additional details regarding the computational requirements of our method, presented in Table 2 and 3 in our main submission. We observe that ATISS requires significantly less time to generate a scene compared to [31, 24]. Note that the computational cost varies depending on the room type, due to the different average number of objects for each room type. Living rooms and dining rooms are typically larger in size, thus more objects need to be generated to cover the empty space. All reported timings are measured on a machine with an NVIDIA GeForce GTX 1080 Ti GPU.

Even though the implementations are not directly comparable, since we cannot guarantee that all have been equally optimized, our findings meet our expectations. Namely, FastSynth [24] requires rendering the scene each time a new object is added, thus it is expected to be significantly slower than both SceneFormer and our model. On the other hand, SceneFormer [31] utilizes four different transformer models for generating the attributes of each object, hence it is expected to be at least four times slower than our model, when generating the same number of objects.

## 6 Perceptual Study

We conducted two paired Amazon Mechanical Turk perceptual studies to evaluate the quality of our generated layouts against FastSynth [24] and SceneFormer [31]. To this end, we first sampled 211 floor plans from the test set and generated 6 scenes per floor plan for each method; no filtering or post-processing was used, and samples were randomly and independently drawn for all methods. Originally, we considered rendering the rooms with the same furniture objects for each floor plan to allow participants to only focus on the layout itself, which is the main focus of our work. However,

# Probability distributions for Desk

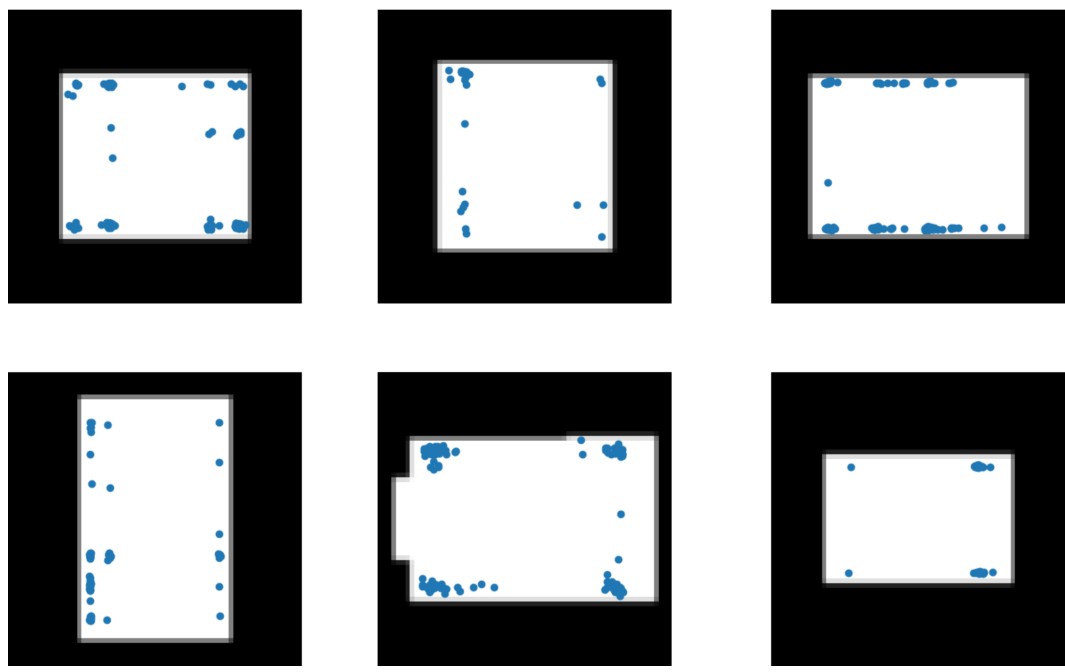

Figure 15: **Location Distributions for Desk.**

# Probability distributions for Nightstand

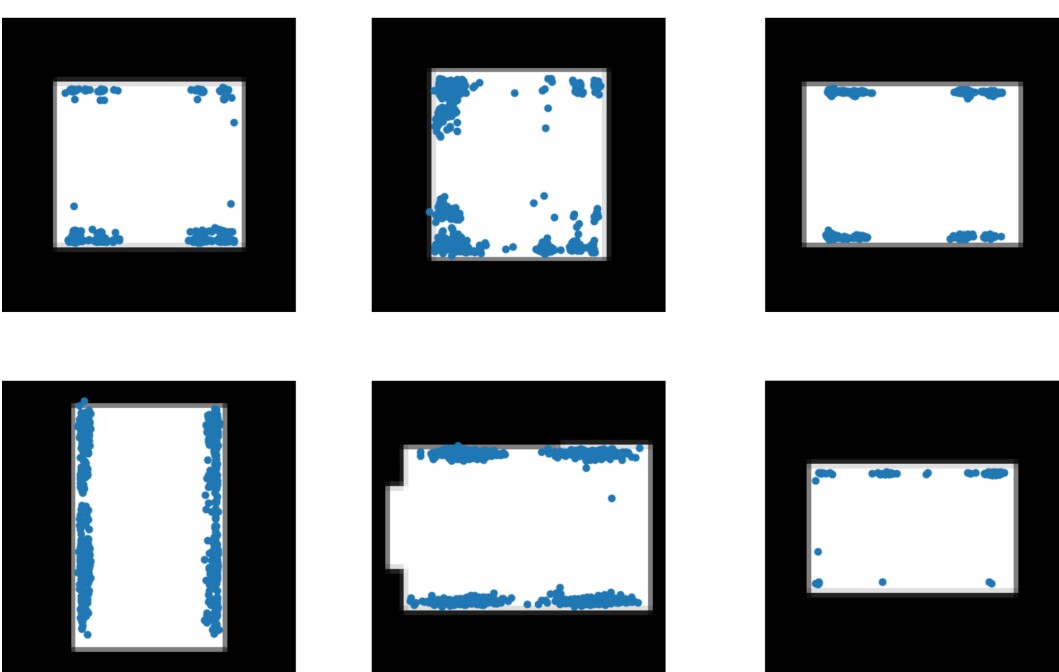

Figure 16: **Location Distributions for Nightstand.**

# Probability distributions for Wardrobe

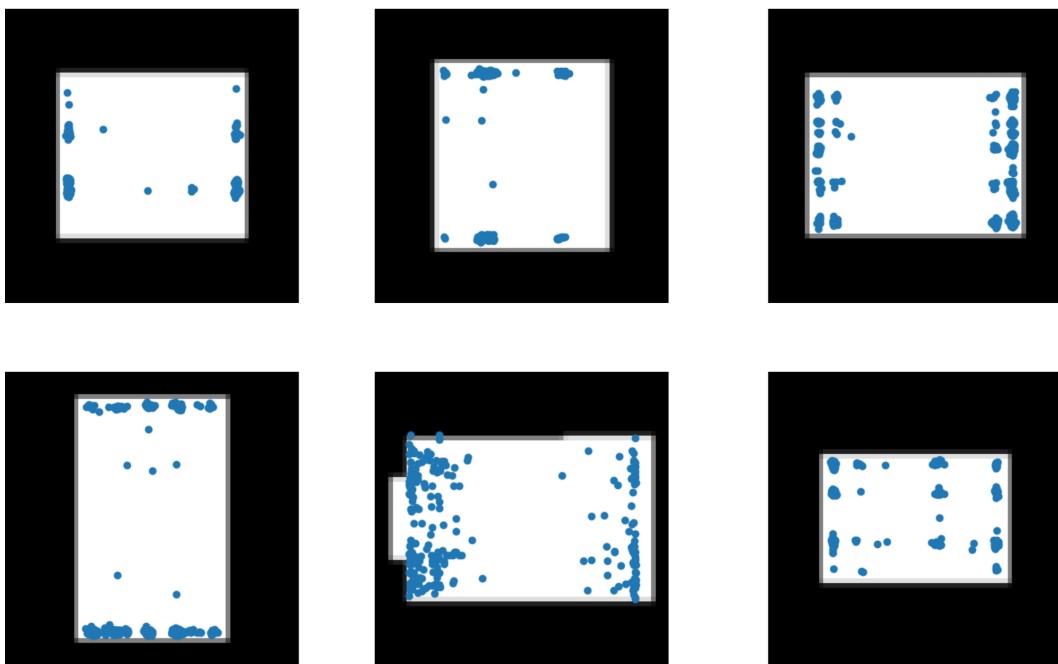

Figure 17: **Location Distributions for Wardrobe.**

since the object retrieval is done based on the object dimensions, rescaling the same furniture piece to fit all predicted dimensions would result in unrealistically deformed pieces that could skew perceptual judgements even more heavily. To avoid having participants focusing on the individual furniture pieces, we added prominent instructions to focus on the layout and **not** the properties of selected objects (see Fig. 18). Each 3D room was rendered as an animated gif using the same camera rotating around the room.

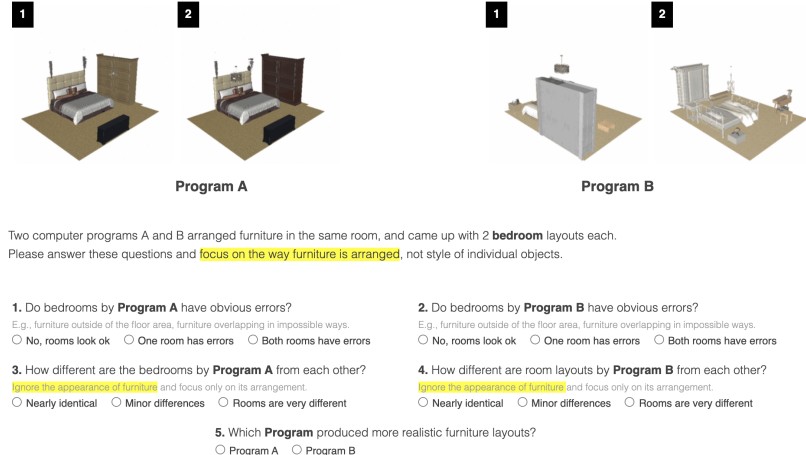

Figure 18: **Perceptual Study UI**. A/B paired questions with rotating 3D scenes (zoom in).

In each user study, users were shown paired question sets: two rooms generated using our method and two generated with the baseline conditioned on the same floor plan. We randomly selected two out of the 6 pre-rendered scenes for the given floor plan, and 5 different workers answered the question set about every floor plan. Namely, the majority of the 6 layouts were shown more than once on average. A / B order was randomized to avoid bias. The question sets posed the same two questions about

scenes generated with program A and B, in order to let users focus on the details of the results and to assess errors of the generated layouts. The last question forced participants to choose between A or B, based on which scene looks more realistic.

Specifically, users were instructed to pay attention to errors like interpenetrating furniture and furniture outside of the floor area and answer if none, one or both layouts for each method had errors. We aggregated these statistics to obtain average error rate per layout, with our method performing nearly twice better than the best baseline [24]. The results on realism in Table 4. in our main submission (first and second row) specify the fraction of the times users chose the baseline over ours. For example, [24] was judged more realistic than ours only 26.9% of the time. Because there was no intermediate option, this means that 73.1% of the time our method was preferred. The last line in Table 4, in our main submission, aggregated preference for our method across both studies.

Workers were compensated $0.05 per question set for a total of USD $106. The participation risks involved only the regular risks associated with the use of a computer.

# 7 Additional Related Work on Indoor Synthesis

In this section, we discuss alternative lines of work on indoor scene synthesis. Fisher et al. [5] propose to represent scenes using relationship graphs that encode spatial and semantic relationships between objects in a scene as well as the identity and semantic classification of each object. Then, they introduce a graph kernel-based scene comparison operator that allows for retrieving similar scenes, performing context-based model search etc. Such representations have been subsequently adopted in models that generate scenes conditioned on user provided constraints and interior design guidelines [16] or rely on a set of example images for generating plausible room layouts [4]. Another line of research [6, 9] leverage activity-associated object relation graphs for generating semantically meaningful object arrangements. Finally, another line of research [1, 15] parses text descriptions into a scene relationship graph that is subsequently used for arranging objects in a 3D scene.

# 8 Discussion and Limitations

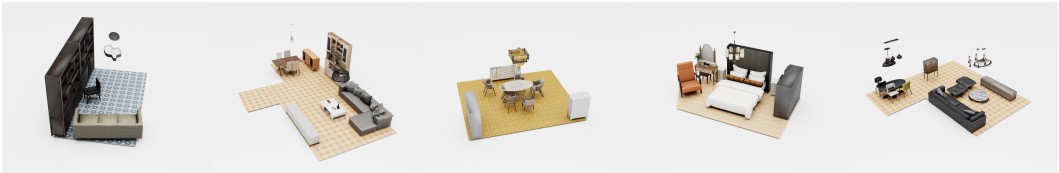

Figure 19: **Failure Cases**. We visualize various failure cases of our model for different toom types.

Lastly, we discuss the limitations of our model and show some examples of failure cases in Fig. 19. One type of failure case that is illustrated in Fig. 19 is overlapping objects, in particular chairs for the case of living rooms and dining rooms (see second and third column in Fig. 19). As we already discussed in Sec. 2, to be able to use the 3D-FRONT dataset, we performed intense filtering to remove objects that intersect with each other. However, we found out that not all problematic arrangements were removed from the dataset, which we hypothesize is the reason for such failure cases. Another type of failure case that we observed, which is also related to the existence of problematic rooms in our training data, is the unnatural orientation of objects (e.g. chair facing the bookshelf in first column of Fig. 19 or chair facing opposite of the table in last column of Fig. 19.) Note that these failure cases are quite rare, as also indicated by our quantitative analysis in Sec. 4.1 in the main submission as well as the perceptual study in Sec. 4.3, but our method does not guarantee error-free layouts and there is room for improvement.

Our approach is currently limited to generating object properties using a specific ordering (category first, followed by location, then orientation and lastly size). To further expand the interactive possibilities of our model, we believe that also the object attributes should be generated in an order invariant fashion, similar to the objects in the scene. Furthermore, in our current formulation, the object retrieval is disconnected from the attribute generation. As a result we cannot guarantee that the retrieved objects would match with existing objects in the scene. To address this, in the future, we

plan to also incorporate style as an additional object attribute to allow for improved object retrieval. Incorporating style information, would also allows us to generate rooms conditioned on a specific style. Another exciting research direction that we would like to explore is combining ATISS with existing compositional representations of objects [27, 21, 17, 22, 20, 18]. This will allow us to generate 3D scenes with control over the object arrangement, object parts and part relationships. Due to the unique characteristics of compositional representations representations, our generated scenes will be fully controllable i.e. it will be possible to manipulate objects and object parts, edit specific parts of the scene etc.

| Scene Layout | Training Sample | FastSynth | SceneFormer | Ours |
|---|---|---|---|---|

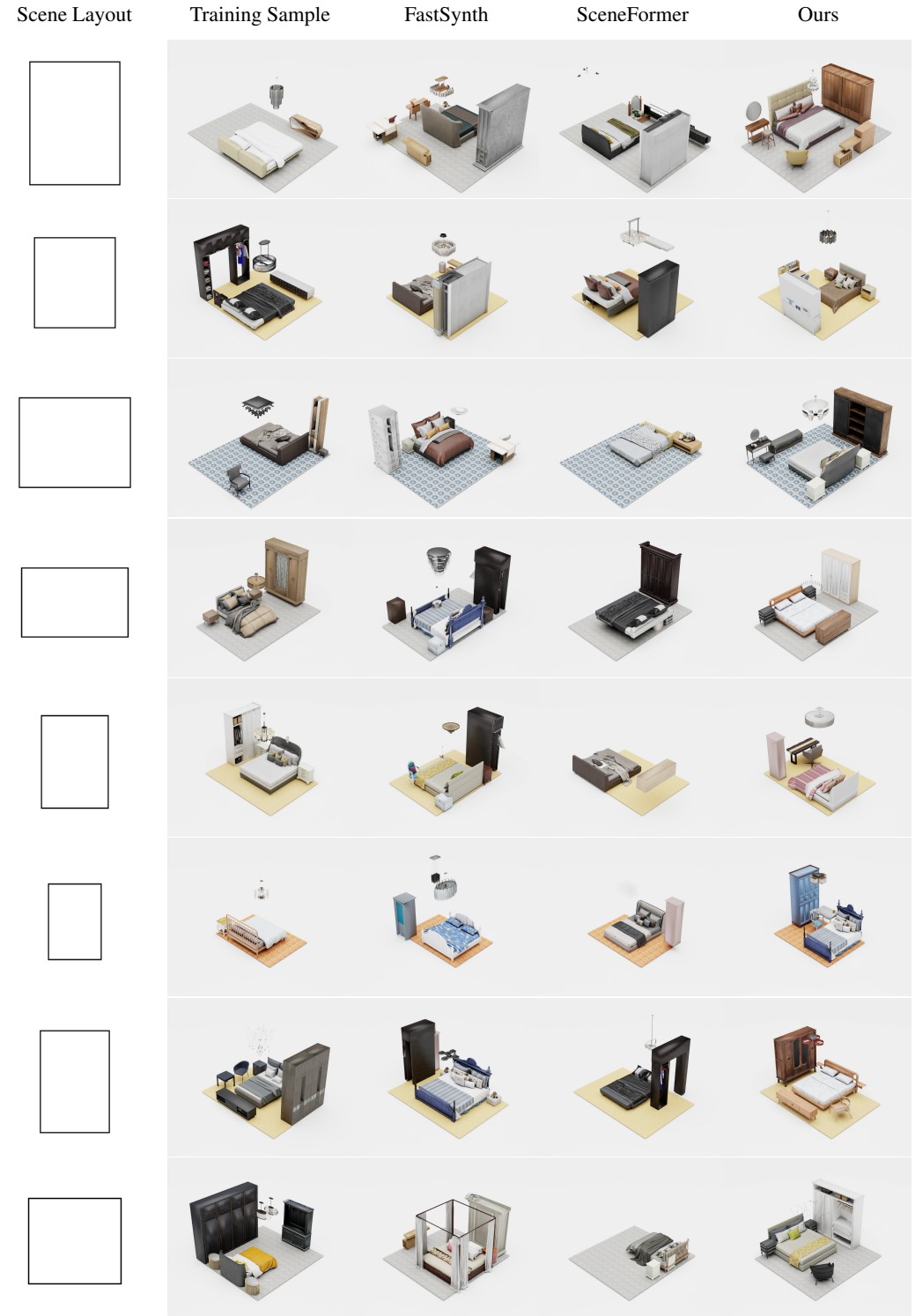

Figure 20: **Qualitative Scene Synthesis Results on Bedrooms**. Generated scenes for bedrooms using FastSynth, SceneFormer and our method. To showcase the generalization abilities of our model we also show the closest scene from the training set (2nd column).

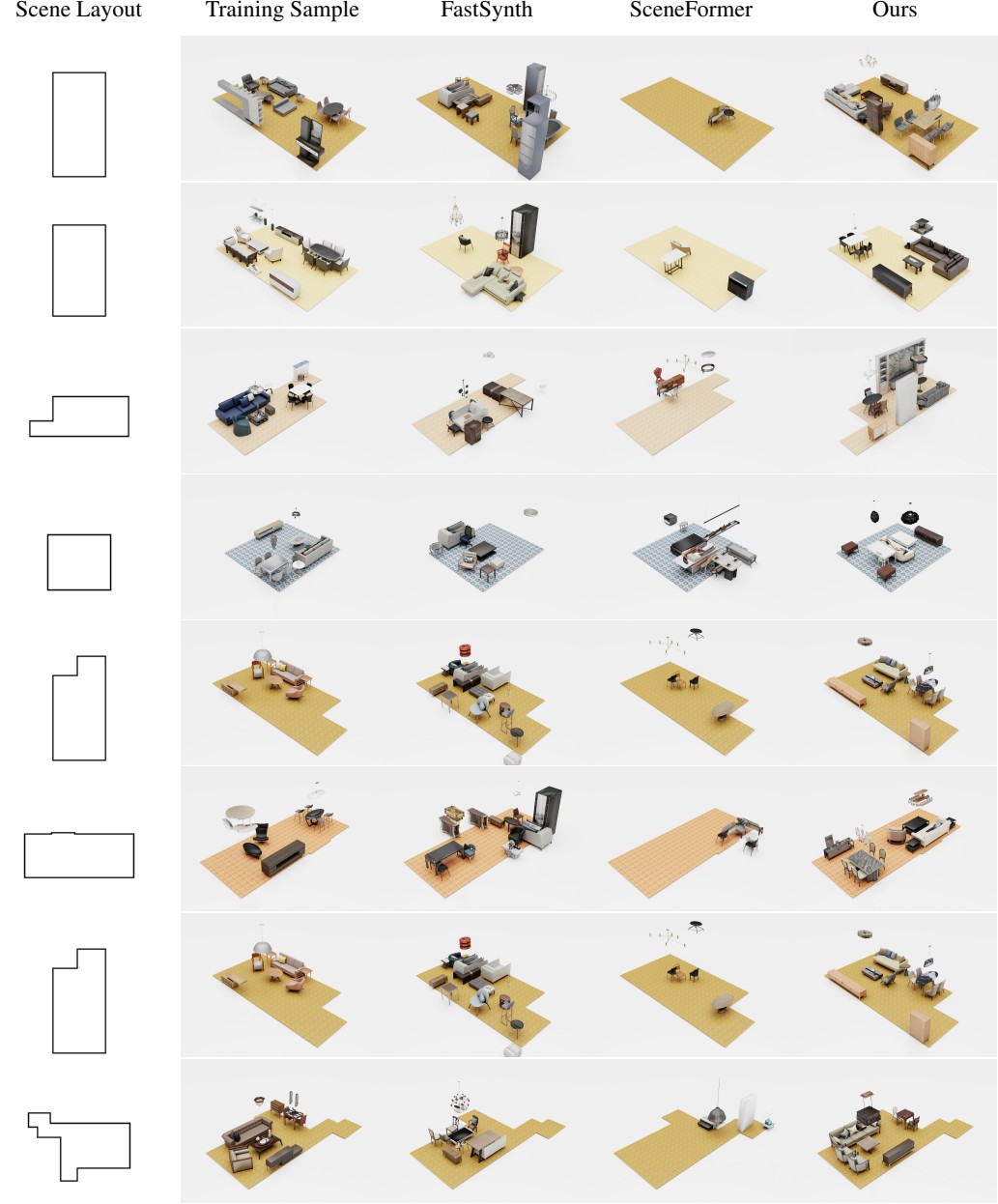

Figure 21: **Qualitative Scene Synthesis Results on Living Rooms**. Generated scenes for living rooms using FastSynth, SceneFormer and our method. To showcase the generalization abilities of our model we also show the closest scene from the training set (2nd column).

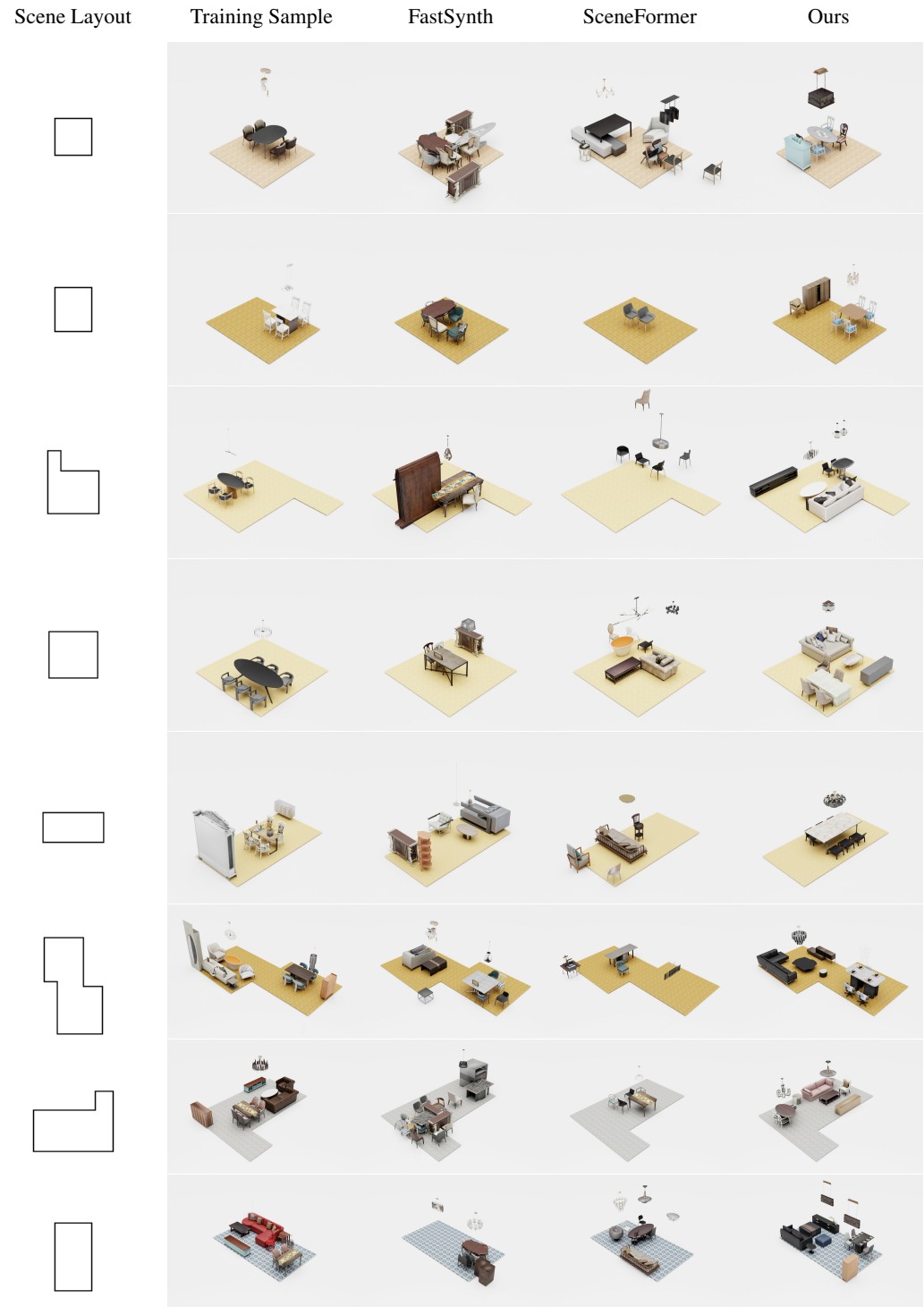

| Scene Layout | Training Sample | FastSynth | SceneFormer | Ours |

Figure 22: **Qualitative Scene Synthesis Results on Dining Rooms**. Generated scenes for dining rooms using FastSynth, SceneFormer and our method. To showcase the generalization abilities of our model we also show the closest scene from the training set (2nd column).

| Scene Layout | Training Sample | FastSynth | SceneFormer | Ours |
|---|---|---|---|---|

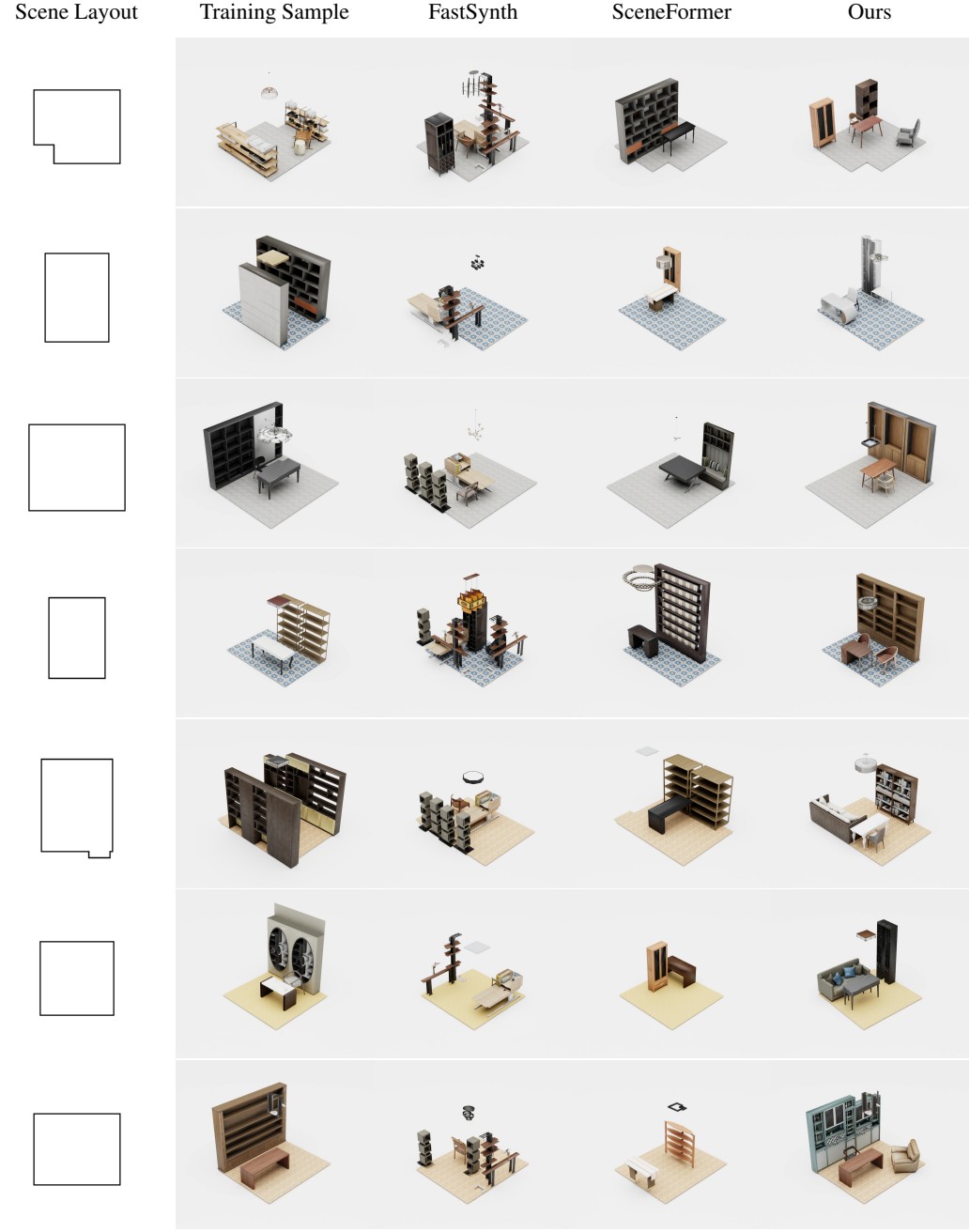

Figure 23: **Qualitative Scene Synthesis Results on Libraries**. Generated scenes for libraries using FastSynth, SceneFormer and our method. To showcase the generalization abilities of our model we also show the closest scene from the training set (2nd column).