# OpenReview forum: "ATISS: Autoregressive Transformers for Indoor Scene Synthesis"
_NeurIPS.cc/2021/Conference — NeurIPS 2021 Poster_

### Official Review · Reviewer_urVi · 2021-07-04

**Rating:** 7
**Confidence:** 3

**Summary:**

The submission proposes a permutation-invariant autoregressive generative model based on transformers for populating 3D rooms with furniture objects.  The model is trained with labeled 3D bounding boxes and is trained and evaluated using the 3D-FRONT dataset.  The proposed model predicts attributes (category, size, location - centroid of bounding box, and orientation - rotation around up axis) for each object in an autoregressive manner using a transformer-based model until stop is predicted.  The category is modeled using a categorical distribution, and the other attributes are modeled using a mixture of logistic distributions.  To model the objects in a scene and a unordered set, the model does not encode the position of the object in the sequence.  To allow the model to handle different permutations of objects, the model is trained with different permutations.  The output prediction of the model is effectively a set of oriented bounding boxes with categories, which are then replaced with an appropriate CAD model with a matching category and closest in size.  Experiments comparing the proposed model against prior work (FastSynth and SceneFormer, both retrained on the 3D-FRONT dataset) shows the proposed model can produce higher quality furniture layouts given a room outline and room type.  There are also qualitative examples of the model used for scene completion, identifying and correcting poor object placement, and suggest objects to place at a given location.  Ablations on the number of logistic distributions, and layout encoder is provided in the supplemental.

Main contributions
- Transformer-based model for indoor scene synthesis that treats the objects as unordered sets during training, and autoregressively generate each object with their attributes
- Set of experiments comparing the proposed model against prior work

**Ethical Concerns:**

There are no ethical concerns.


**Limitations And Societal Impact:**

Limitations of the method is described in the submission.  There is not much discussion of how the limitations would translate to negative societal impact.


**Main Review:**

Originality: While idea of using transformers is not new (it has been explored in the unpublished work SceneFormer), this work provides convincing experiments showing the proposed model outperforms prior work on indoor 3D scene synthesis.  It also point out that while the scene synthesis can be formulated as a sequence generation problem, it is important to allow for different ordering during training as applications such as scene completion may require generation from a random set of objects.

Quality: The submission appears to be technically sound with reasonable experimental setup and results (see weaknesses for missing comparisons).

Clarity: The paper is well written and clear.  When describing various parts of the model, the rationale for choosing certain design decision is provided.

Significance: The ability to rearrange furniture in realistic layouts will allow for automatic generation of indoor scene data at scale.  This work is a step toward that direction.  It is also one of the first works to move away from the SUNCG dataset which is no longer available for download.

Strengths:
- The proposed model shows how to use transformers to model scenes as unordered sets of objects
- The experiments are thorough with both automated evaluation and perceptual study with users.
- Experiments show that the proposed model generates higher-quality rooms and is faster than prior work

Weaknesses:
- The train/test split used is unclear (see "Other comments and questions")
- There is limited ablation studies
- Some of the claims made in the paper is inaccurate (see "Other comments and questions")


Missing References:
The submission is missing discussion of older/alternative lines of work on indoor scene synthesis.  Some examples are provided below:
- Interactive Furniture Layout Using Interior Design Guidelines, Merrell et al, SIGGRAPH 2011
- Example-based synthesis of 3D object arrangements, Fisher et al, SIGGRAPH ASIA 2012
- Learning spatial knowledge for text to 3D scene generation, Chang et al, EMNLP 2014
- Language-driven synthesis of 3d scenes from scene databases, Ma et al, SIGGRAPH Asia 2018
- Activity-centric Scene Synthesis for Functional 3D Scene Modeling, Fisher et al, SIGGRAPH Asia 2015
- Adaptive synthesis of indoor scenes via activity-associated object relation graphs, Fu et al, SIGGRAPH Asia 2017
- Fast and Scalable Position-Based Layout Synthesis, Weiss et al, Transactions on visualization and computer graphics 2018

The references should also be proofread for errors:
- Citation for Metropolis procedural modeling is duplicated ([50,51])
- Ideally when a preprint is superseded by a refereed publication at a conference/journal, the citation should be for the refereed publication (not the preprint).


Other comments and questions:
- Equation 6: Should the R^M be R^C since it corresponds to the c (category of object j)?

- The notation for the per-object contextual embedding C_j is a bit confusing.  It is unclear if the C_j refers to the 64-dimensional embedding (as in supplement line 32), or to the embedded attributes (as suggested by the main paper Lines 151,159) before it is projected to a 64-dimensional vector.

- Lines 242-249: The description for Scene Completion states that both the baseline methods [45, 58] fails to generate any objects in Figure 7.  This is not true as SceneFormer [58] has clearly added some objects to both scenes.

- Line 197: I'm not sure that the 3D-FRONT dataset is "professionally designed".  The scenes from 3D-FRONT are based on a set of scenes designed using the Alibaba Topping Homestyler tool (it's unclear if the users were professional or just regular users), and due to copyright restriction, the CAD models in the scenes are algorithmically replaced/removed.  So the final dataset could not be said to be "professionally designed".

- The split of scenes into training and test set is not really specified.  Line 270 mention there is 211 test set plans, but it is not that clear where this split comes from and how is it split.  Is this a standard split from the 3D-FRONT dataset?  Is the split done so that the rooms for the training and test set come from different houses?  Is the set of objects shared between the splits?  Is there a validation split for hyperparameter tuning?

- Since the results for Table 1 are for 10 runs, it would be good to report the error bars for those.

- The paper points out that one limitation of prior work is that they are trained on a sorted sequence of objects and thus fails to fail to generate any objects giving an arbitrary set of objects as input. It's not actually clear to me why the FastSynth[45] model requires a ordering during training.  According to the FastSynth[45] paper, it is trained by randomly removing a set of objects in the scene but for some reason they require a stable ordering because otherwise "there are too many valid possible categories at each step".  Do you think that a version of the FastSynth model that is trained with random removal of objects (without the ordering) would be able to generate addition objects for the cases presented in Figure 7?

**Time Spent Reviewing:**

4.5

---

> ### Author Response · Authors · 2021-08-10
> **Answer to Reviewer uRVi**
>
> Thank you for your review. For the final version, we will discuss the additional related work brought up in your review and incorporate your comments for Lines 242-249 and Lines 197.
>
> **Q1: The train/test split used is unclear**
>
> We split the preprocessed rooms such that $70$% is used for training, $20$% for testing and $10$% for validation. Note that the 3D-FRONT dataset comprises multiple houses that may contain the same room, e.g the exact same object arrangement might appear in multiple houses.  Thus splitting train and test scenes solely based on whether they belong to different houses could result in the same room appearing both in train and test scenes. Therefore, we don't simply randomly select rooms from houses but we select from the set of rooms with distinct object arrangements.  We will clarify this in the final version. Moreover, should the paper be accepted, we intend to release the ids of the scenes we used for training, testing and validation to ensure fair comparison with follow-up works.
>
> Regarding hyperparameter tuning, indeed we used the validation set to decide
> the values of various parameters of our model such as the number of mixture of
> logistics, the dimensionality of various feature representations etc.
>
> **Q2: Equation 6: Should the R^M be R^C since it corresponds to the C (category of object j)?**
>
> Thanks for spotting this typo. It should indeed be $\mathbb{R}^C$ instead of $\mathbb{R}^M$, where $C$ is the total number of object categories in the dataset. We will fix this for the final version. Thank you.
>
> **Q3: The notation for the per-object contextual embedding C_j is a bit confusing.**
>
> We will make the notation consistent between the main paper and the supplementary to avoid confusion. Thank you for pointing this out.
>
> **Q4: Since the results for Table 1 are for 10 runs, it would be good to report the error bars for those.**
>
> Since Table 1 was quite dense, we report the corresponding standard deviations in Table 4 in the supplementary. We are happy to move these results to the main paper.
>
> **Q5: Do you think that a version of the FastSynth model that is trained with
> random removal of objects (without the ordering) would be able to generate
> additional objects for the cases presented in Figure 7?**
>
> While it is difficult to explain why FastSynth struggles when trained without a stable ordering, a possible explanation for this could be that in general maintaining a 2D state is inherently suboptimal for the task of predicting the category of the next object to be added in the scene. This can be justified by the fact that the model needs to extract several object attributes from the 2D scene representation as well as higher level concepts such as object frequency. This is also why it is probably necessary to provide the category count along with the scene representation. Using a stable ordering makes the task significantly easier since it can be solved by looking just at the category count and if there is empty space in the image representation. Regarding Fig. 7, we believe that training FastSynth with random ordering should be able to place objects in the scene, however based on the observation of the authors of [1] the final scene would probably be less coherent.

---

### Official Review · Reviewer_DWcp · 2021-07-17

**Rating:** 7
**Confidence:** 4

**Summary:**

This paper proposes an transformer based framework for autoregressive indoor scene synthesis. A scene is modelled by an orthographic projection of the floor geometry (encoded by a CNN), and an unordered set of objects, characterized by category (encoded with a learned embedding), location, orientation and size (encoded with positional encoding). These attributes are then fed into a transformer encoder, which predicts a feature vector that is then decoded by a cascade of MLPs into attributes of the object to be predicted, where a categorical distribution is used for category, and a mixture of logistics is used for the remaining three attributes.

The proposed model is thus able to generate indoor scenes by autoregressively add the objects in arbitrary order. The scene synthesis results outperform relevant prior works convincingly with respect to multiple evaluation metrics. The paper also showcases a myriad of applications such as scene completion, failure detection and object suggestion, which are enabled by the design of the method.

**Limitations And Societal Impact:**

Not that I am aware of.

**Main Review:**

I like this paper overall. The authors demonstrate clear knowledge of relevant works of scene synthesis and works that apply transformers to computer graphics. The design choices make intuitive sense and suits the task well e.g. the use of positional encoding for continuous input attributes and the use of mixture of logistics for predicting these attributes. The quality of the outputs are impressive: not just in terms of numbers (these can be hard to judge for such generative models), but also in terms of qualitative examples. The output of the proposed method seems to be handling challenging cases (which, as far as I know, are not handled by prior works well) pretty convincingly e.g. symmetrical chairs around dining tables, placement of sofa around coffee tables, etc. It is also very impressive that the proposed method can avoid collisions without explicit collision handling. While I do think the baselines are not retrained particularly well, judging from the results from the original works, the results still appear to be more convincing.

The most outstanding limitation of this work has to do with novelty: the autoregressive scene generation formulation is similar to multiple prior works. The claims that this work is novel in terms of being able to handle random ordering and being able to train everything in one model, to me, is not particular convincing. The use of transformers for generating 2D/3D outputs, again, is not a new concept, which the authors also acknowledge themselves.

I cannot be more positive on this paper primarily due to the novelty concerns. Still, I think this is a very well engineered piece of work and clearly advances state of the art (though on a topic traditionally more of interest in graphics/vision community), and can inspire future works both in 3D generative models and in using transformer for autoregressive generation. I lean towards acceptance, but won't be too bothered if others decide against this paper.

Additional comments and questions below.

=====Method=====

-Orientation can be ambiguous for symmetrical objects. Will that cause problems with training, or do the output distribution naturally handles that?

-Also for orientation: how is SO(2) mapped to R? Are all orientations mapped onto [0, 2pi) or can they fall anywhere? Subsequently, how do the mixture of logistics handle the "wrap-around" for decoding?

-Why predicting the 3D cuboid centroid instead of the 2D one? It appears to me that predicting the 3D centroid will hurt disentanglement between location & size, since 2location_y = size_y for all ground-attached objects? What is the behavior in practice, and do the network consistently put everything on ground level?

-I wonder if the current pipeline can be naturally extended to scenes with hierarchy e.g. objects on desks.

-What information is carried in the transformer output q hat? Is it carrying a summary of the current partial scene or is it carrying the "object feature" as claimed in the paper? It appears to me that if the latter is the case, then this vector need to carry a lot of information for it to handle all these distributions when the sampled next object can differ in category/location/orientation? Any intuitions on this?

=====Results & Evaluation=====

-I am very curious of the behavior of the proposed model in Figure 6, bottom left. It seems that it is learn to match the orientation of the objects with boundaries rarely seen in the training set? Does this work for all wall orientations or is it something a bit more specific to this input?

-I think figure 7 is not fair. Baselines trained on fixed sequences certainly won't be able to handle such cases. A fairer comparison would be to compare this against versions of the baselines that are trained on random orderings, which they should be able to do.

-I would like to see some visualizations of the distributions predicted, in addition to full generated scenes. This will help judging if the model is overfitting to specific location/orientation/size or is it learning something that is more general. I get that figure 5 partially showcases diversity, but this might be due to difference in generation order, not due to the model capture all possible modes at every single step. A bit more evaluations on this would be nice.

-Would also like to see just a bit more qualitative results (sampled randomly), perhaps in the supplementary?

=====Post-Rebuttal Comment=====

I thank the authors for their details response to my concerns. I am convinced by most of them, though there are a few that I am still left unconvinced with, which I list below. I have raised my score from 6 to 7 in response to the rebuttals, and the fact that other reviewers do not express too much concerns regarding novelty.

- Ordering & Novelty: I still don't feel that it is the best idea to claim random ordering as a main novelty - in the case of short sequences it isn't really that challenging to do, and there's plenty of prior works that handle much more challenging cases of set generation e.g. [53]. While I do agree that it is nice that this model can do autoregressive scene synthesis with random ordering, personally I think the outstanding quality of results is the more impressive part.
- Compare against prior works on ordering (Q8 &  uRVi Q5): If I understand it correctly, it should be easy to train a version of the existing works with random sequences of objects. It should be a relatively straightforward modification with their data loader. I don't doubt that they will perform worse with this modification, but just feel that adding it will make the claims in the paper much clearer and remove the need for indirect speculations.
- Predicted distributions (Q9): I think there's quite a bit of overfitting going on, but I guess this is OK as the emergent final layouts are indeed diverse. Still, this might be worth discussing a bit in the paper and might serve as a good direction for future works. Besides, some visualizations with a partial scene instead of an empty room might be more illustrative.

**Time Spent Reviewing:**

3

---

> ### Author Response · Authors · 2021-08-10
> **Answer to Reviewer DWcp**
>
> Thank you for your review.
>
> **Q1: Novelty Concern:**
>
> We would like to clarify that one of the key contributions of our work is the unordered set generation (via an autoregressive transformer), which we experimentally demonstrate to be important for high quality and interactive furniture layout synthesis. To the best of our knowledge, neither of the above was explored in existing work. Also note that while autoregressive scene generation pipelines have indeed been explored in prior work, they were only able to handle ordered sequences of objects. Modifying existing pipelines to handle random orderings is not trivial. For example, for the case of [1], the authors observed that not imposing a stable canonical ordering resulted in incoherent scenes across multiple object insertions.
>
>
> **Q2: Orientation can be ambiguous for symmetrical objects. Will that cause problems with training, or does the output distribution naturally handle that?**
>
> In general, we note that our model accurately captures the orientation distributions found in the dataset. If symmetrical objects have random orientations, then the predicted distribution should be close to uniform. For example, for the case of lamps, which are the most common symmetrical type of object in the dataset, the orientation in the training data is not random but usually $0,90,180$ degrees etc. Our model learns to predict these modes.
>
>
> **Q3: Also for orientation: how is SO(2) mapped to R? Are all orientations mapped onto [0, 2pi) or can they fall anywhere? Subsequently, how do the mixture of logistics handle the "wrap-around" for decoding?**
>
> Both the target and the predicted orientations can take values in the range $[-\pi, \pi]$. The sampled values are clamped to this range similar to [3, 4]. Note that the mixture of logistics loss takes into account this clamping as discussed in [3, 4]. In general, to compute the negative log-likelihood for a value for the discretized mixture of logistics loss we split the range of possible values to $N$ equal segments except for the segments at the boundaries that also contain the tails of the distribution. When computing the probability we take into account those tails, namely we assign the probability from $-\infty$ to the boundary to the first segment and from the boundary to $+\infty$ to the last segment. We will clarify this in the revision. For more details please refer to [3, 4].
>
>
> **Q4: Why predicting the 3D cuboid centroid instead of the 2D one? It appears to
> me that predicting the 3D centroid will hurt disentanglement between location &
> size, since 2location_y = size_y for all ground-attached objects? What is the
> behavior in practice, and do the network consistently put everything on ground
> level?**
>
> Thank you for bringing up this point. Indeed, if we were to model the location of objects using the 3D centroid of the bottom face of its bounding box, for the majority of objects its $y$ coordinate would be 0. We also considered this alternative parametrization, because similarly, we were hypothesizing that such parametrization could disentangle location and size and potentially also facilitate learning since the $y$-axis is predominantly 0. However, when we compared the two parametrizations in our experiments, we did not notice any significant difference either in the training behaviour of our model (e.g. faster training), smoother convergence, or in the quality of the generated scenes. We thus decided to model the location using the 3D centroid of the bounding box.
>
> Moreover, in all our experiments, we observed that apart from lamps, our model
> was consistently placing all objects on the ground level.
>
> We will provide these additional details in the final version to avoid confusion. Thank you for raising the point.
>
> **Q5: I wonder if the current pipeline can be naturally extended to scenes with
> hierarchy e.g. objects on desks.**
>
> We are also very interested in extending our work on more higher-level concepts such hierarchies in the form of support or functional relationships. Given suitable training data that provide this kind of annotations, our model could be extended to also predict such object-level relationships. For instance, a simple dot product of $\hat{q}$ with all $\hat{\mathbf{C}_j}$ could be used for predicting the relationship of the newly added object with the objects already present in the scene.
>
> **Q6: What information is carried in the transformer output q hat?**
>
> It is generally hard to state what information is captured by $\hat{q}$. Our intuition is that it encodes information for the next object to be added in the scene. Taking into consideration that we use a relatively large transformer architecture (4 transformer layers with 8 attention heads) and very small prediction heads in the attribute extractor (one or two fully connected layers), we believe that the information is predicted by the transformer and simply decoded in the attribute extractor. Hence, $\hat{q}$ should contain information about the object and not the scene.
>
>
> **Q7: I am very curious of the behavior of the proposed model in Figure 6, bottom
> left. It seems that it is learn to match the orientation of the objects with
> boundaries rarely seen in the training set? Does this work for all wall
> orientations or is it something a bit more specific to this input?**
>
> In general, we observe that our model consistently places the objects within
> the room boundaries and aligns them with the walls. There is nothing
> specific to this particular input since the same behaviour can also be observed in all our
> generated scenes. Note that this behaviour can also be observed for FastSynth
> [1], despite the fact that some objects have implausible orientations.
>
>
> **Q8: I think figure 7 is not fair. Baselines trained on fixed sequences
> certainly won't be able to handle such cases. A fairer comparison would be to
> compare this against versions of the baselines that are trained on random
> orderings, which they should be able to do.**
>
> We agree that models trained with fixed orderings such as [1,2] would not be able to complete these scenes successfully.  However, our intention for the experiment of Figure 7 was to showcase the importance of the permutation invariance property in scene generation. In particular, we wanted to demonstrate that models that consider ordering based on the frequency of objects cannot handle cases where less common objects appear before more frequent objects. We will clarify the text to better reflect our intention.
>
> To ensure a fair comparison to FastSynth [1] and SceneFormer [2], we consider a variant of our model with fixed ordering (see Ours+Order in Table 1) and show that even this variant consistently outperforms both baselines in terms of the FID score and the classification accuracy for all room types while also performing better in terms of the KL Divergence for the libraries and the dining rooms. We will clarify this further in the paper.
>
> **Q9: Additional visualizations of the predicted distributions**
>
> This is a great suggestion. Upon your request, we randomly selected 6 floor plans and generated 500 rooms for each floor plan. Subsequently, we created scatter plots for the locations of various object categories, which you can find in the following anonymous link https://imgur.com/a/brofjNg. We observe that the location distributions cover all meaningful areas of the rooms, which is also why we see diverse results in Fig 5, in our main submission. We will include such additional visualizations in the final version of the paper.
>
>
> **Q10: Additional qualitatively results (sampled randomly)**
>
> We would like to clarify that we provide additional qualitative results,
> sampled randomly, for all room types in the supplementary in Fig. 11,12,13,14.
>
>
> [1] Daniel Ritchie, Kai Wang, and Yu-An Lin. Fast and flexible indoor scene synthesis via deep convolutional
> generative models., CVPR , 2019.
>
> [2] Xinpeng Wang, Chandan Yeshwanth, and Matthias Nießner. Sceneformer: Indoor scene generation with
> transformers. ARXIV 2020
>
> [3] Tim Salimans, Andrej Karpathy, Xi Chen, and Diederik P. Kingma. Pixelcnn++: Improving the pixelcnn
> with discretized logistic mixture likelihood and other modifications., ICLR 2017.
>
> [4] Aäron van den Oord, Sander Dieleman, Heiga Zen, Karen Simonyan, Oriol Vinyals, Alex Graves, Nal
> Kalchbrenner, Andrew W. Senior, and Koray Kavukcuoglu. Wavenet: A generative model for raw audio., ARXIV 2016.

---

### Official Review · Reviewer_z3Jv · 2021-07-17

**Rating:** 8
**Confidence:** 4

**Summary:**

The paper proposes an approach for 3D scene synthesis based on a transformer architecture.  The approach encodes 3D scenes from a top-down floorplan image and a set of per-object learned embeddings that capture the category, size, location, and orientation of each object.  Critically, the formulation addresses the permutation-invariant nature of object sets in 3D scenes, in contrast with prior work which relied on generation of scenes through an ordered sequence of object insertions.  The proposed approach is demonstrated on three tasks: 3D scene synthesis "from scratch", scene completion (i.e. conditioning synthesis on a partial scene), suggestion of objects to place at a particular location in a scene, and detection of implausibly placed objects.

The experiments contrast the proposed approach against baselines from prior work, as well as an ablation that requires object ordering.  Scenes generated using the approach are shown to be of higher quality (as measured by FID, classification, and KL divergence metrics) relative to scenes generated using the baseline methods.  Additional analysis experiments show that the approach also generalizes well to out-of-domain floorplan shapes, and that it is significantly faster than prior work.  Finally, a perceptual study carried out on Amazon Mechanical Turk shows that scenes generated by the approach are deemed to contain significantly fewer object placement errors and to be more realistic that scenes generated by the baselines.

**Limitations And Societal Impact:**

The authors discuss some directions for future work and connect them limitations of the proposed approach (in the main paper, and in the supplementary material).  Societal impacts of the work are mentioned briefly in the conclusion to the paper.

**Main Review:**

Originality:
The paper's use of a transformer architecture with a permutation-invariant object set encoding is novel, and sets it apart from prior work.
 On the other hand, the discussion of related work on graph-based scene synthesis is missing early work that introduced graph-based representations for 3D scenes (​"Characterizing Structural Relationships in Scenes using Graph Kernels" [Fisher et al. 2011]), and work that adopted such representations for 3D scene synthesis ("Interactive furniture layout using interior design guidelines" [Merrell et al. 2011], "Make it home: automatic optimization of furniture arrangement" [Yu et al. 2011], "Example-based Synthesis of 3D Object Arrangements" [Fisher et al. 2012]).  I would recommend that the authors incorporate and discuss these references as well.

Quality:
The submission is technically sound, as far as I can tell.  The experiments are fairly thorough and support the claims well.  In particular, the attention to detail in clarifying the permutation-invariant nature of the scene synthesis problem statement is valuable.  The careful analysis of of model capacity differences (network parameter comparison) is also quite valuable, and will hopefully inform future work.

Clarity:
The paper is clearly written overall.  There are a few minor low-level writing issues that can be fixed through a proofreading pass.  A specific example that caught my eye in L129-L130: the order of the category, size, orientation, and location is not consistent with the order of the symbols immediately following the description.  Also "logistics distributions" should be "logistic distributions".

Significance:
I believe this paper is likely to inspire follow up work in the area of 3D scene synthesis.  This is a solid paper, and I am in favor of acceptance.

**Time Spent Reviewing:**

4

---

> ### Author Response · Authors · 2021-08-10
> **Answer to Reviewer z3Jv**
>
> Thank you for your positive review. For the final version, we will fix all typos and discuss the additional related work mentioned in your review.

---

### Decision · Program_Chairs · 2021-09-27

**Decision:**

Accept (Poster)

**Comment:**

Reviewers agree, and I concur, that this paper is solid. It has a well executed, relatively scaled up model and idea with good results. I also agree with the relative lack of novelty or insight, so I recommend acceptance as a poster.